# Central role of Prominin-1 in lipid rafts during liver regeneration

Myeong-Suk Bahn[1,4], Dong-Min Yu[1,4], Myoungwoo Lee [1], Sung-Je Jo[1], Ji-Won Lee[1], Ho-Chul Kim[1], Hyun Lee[1], Hong Lim Kim[2], Arum Kim[1], Jeong-Ho Hong [1], Jun Seok Kim[1], Seung-Hoi Koo [1], Jae-Seon Lee [3] & Young-Gyu Ko [1] ✉

Prominin-1, a lipid raft protein, is required for maintaining cancer stem cell properties in hepatocarcinoma cell lines, but its physiological roles in the liver have not been well studied. Here, we investigate the role of Prominin-1 in lipid rafts during liver regeneration and show that expression of Prominin-1 increases after 2/3 partial hepatectomy or CCl₄ injection. Hepatocyte proliferation and liver regeneration are attenuated in liver-specific Prominin-1 knockout mice compared to wild-type mice. Detailed mechanistic studies reveal that Prominin-1 interacts with the interleukin-6 signal transducer glycoprotein 130, confining it to lipid rafts so that STAT3 signaling by IL-6 is effectively activated. The overexpression of the glycosylphosphatidylinositol-anchored first extracellular domain of Prominin-1, which is the domain that binds to GP130, rescued the proliferation of hepatocytes and liver regeneration in liver-specific Prominin-1 knockout mice. In summary, Prominin-1 is upregulated in hepatocytes during liver regeneration where it recruits GP130 into lipid rafts and activates the IL6-GP130-STAT3 axis, suggesting that Prominin-1 might be a promising target for therapeutic applications in liver transplantation.

The liver is a pivotal organ for maintaining homeostasis by regulating metabolism, drug detoxification and bile transportation[1]. Hepatocytes, the major parenchymal cells in the liver, could be damaged by various factors such as surgical operation, alcohol, virus and chemicals, which leads to a decrease in liver mass[2]. To maintain homeostasis, the liver has a unique capability to recover its original mass[3,4]. Many studies for cytokines or growth factors have tried to contribute therapeutic approaches to promote liver regeneration. For example, there are antibody, agonist or antagonist therapy targeting for specific signaling pathway in liver regeneration[5]. In addition, some studies have attempted to use a cell population with stemness for a cell therapy[6,7]. Understanding the molecular mechanisms in liver regeneration is important for application in the field of liver disease therapy.

A 2/3 partial hepatectomy (PHx) is a well-characterized experimental model for liver regeneration in rodents. Mice recover most of their liver mass 7 days after PHx[8]. During liver regeneration, quiescent hepatocytes proliferate by several cytokines and growth factors, such as epidermal growth factor (EGF), hepatocyte growth factor (HGF), fibroblast growth factor (FGF), tumor necrosis factor-α (TNFα), and interleukin-6 (IL-6)[3,9].

IL-6 is a pleiotropic cytokine in the body. After PHx or other liver injuries, gut-derived factors such as lipopolysaccharides (LPS) activate Kupffer cells and resident liver macrophages to secrete IL-6[10]. Secreted IL-6 binds to the interleukin-6 receptor (IL-6R) and then forms a signaling complex consisting of IL-6R and interleukin-6 signal transducer glycoprotein 130 (GP130) in hepatocytes[11]. The complex initiates

[1]Division of Life Sciences, Korea University, Seoul 02841, Korea. [2]Laboratory of Electron Microscope, Integrative Research Support Center, College of Medicine, The Catholic University of Korea, Seoul, Korea. [3]Research Center for Controlling Intercellular Communication, College of Medicine, Inha University, Incheon 22212, Korea. [4]These authors contributed equally: Myeong-Suk Bahn, Dong-Min Yu. ✉e-mail: ygko@korea.ac.kr

several downstream signaling pathways, including Janus kinases (JAKs), signal transducer and activator of transcription 3 (STAT3), MAP kinases and the PI3 kinase pathway.

*Il-6* knockout impairs hepatocyte proliferation and induces liver necrosis after PHx in mice, preventing liver mass recovery. As a result, *Il-6* knockout significantly increases mortality after surgery. Thus, a single injection of IL-6 rescues this phenotype in *Il-6* knockout mice[12]. In addition, liver-specific *Stat3* knockout impairs the DNA synthetic response in hepatocytes and decreases the expression of $G_1$ phase cyclins such as cyclin D1 and cyclin E[13]. Consistent with the important role of the IL-6 signaling pathway during liver regeneration, liver-specific knockout of suppressor of cytokine signaling 3 (SOCS3), a negative regulator of the STAT3 pathway, exhibits prolonged activation of STAT3 and enhances hepatocyte proliferation, resulting in accelerated liver mass replenishment after PHx[14].

Prominin-1 (PROM1), also known as CD133, is a penta-span transmembrane glycoprotein. PROM1 is associated with distinct detergent-resistant lipid rafts[15] and is found in membrane protrusions such as filopodia and microvilli[16]. PROM1 has been studied as one of the most widely used cancer stem cell (CSC) markers in various human tumors, including the liver[17–20]. In addition to cancer stem cells, PROM1 is also expressed in normal stem cells, including hematopoietic stem cells and various epithelial cells, in the brain, kidney, digestive track, and liver[21–23]. Specifically, PROM1 has been known to express in the liver (human and mouse), including canals of Hering, bile ducts and hepatocytes[24,25]. Indeed, PROM1 regulates the glucagon and TGF-β signaling pathways in the liver by interacting with radixin and SMAD7, respectively[26,27].

Because PROM1, a marker for hepatic progenitor cells, is also upregulated in hepatocytes after liver injury[27], the upregulated PROM1 might regulate various signaling pathways related to hepatocyte proliferation. Here, we observed a significant increase in the expression of PROM1 in hepatocytes during liver regeneration after PHx or $CCl_4$ injection. Liver-specific *Prom1* knockout (*Prom1^LKO*) mice showed impaired liver regeneration because of reduced hepatocyte proliferation. Mechanistically, we found that the increased PROM1 in hepatocytes confined GP130 to lipid rafts and facilitated activation of STAT3. These results demonstrated that PROM1 plays an important role during liver regeneration through the IL6-GP130-STAT3 signaling pathway.

## Results

### PROM1 is upregulated in hepatocytes during liver regeneration

To investigate the expression of PROM1 during liver regeneration, we performed 2/3 partial hepatectomy (PHx) in wild-type mice. We found that the mRNA level of PROM1 increased after PHx by qRT-PCR (Fig. 1a). The mRNA level of PROM1 peaked 48 hours after PHx and then gradually decreased. Consistently, immunoblotting confirmed that the protein level of PROM1 increased 24, 48 and 120 hours after PHx (Fig. 1b and Supplementary Fig. 1A). Next, we determined which cells expressed PROM1 in the liver by PROM1 double immunofluorescence with hepatocyte nuclear factor 4α (HNF4α as a specific marker of hepatocytes) or cytokeratin-19 (CK19 as a specific marker of ductal cells) (Fig. 1c, d). PROM1 was mainly expressed in ductal cells of sham liver, whereas it was expressed in hepatocytes of PHx liver. In our previous report26, we confirmed that PROM1 was expressed in microvilli of primary hepatocytes using an electron microscopy. Thus, we tried to confirm the localization of PROM1 in the microvilli of hepatocytes in PHx liver by correlative light and electron microscopy (CLEM). As shown in Supplementary Fig. 1B, immunogold-labeled PROM1 was localized in the microvilli of hepatocytes in wild-type (Prom1f/f) liver, but not in Prom1LKO liver. These data showed that PROM1 was localized in the microvilli of hepatocytes from PHx liver.

To further clarify the cell types expressing PROM1 during liver regeneration, we generated a lineage-tracing mouse in which tdTomato (tdTom) was expressed by tamoxifen in PROM1-positive cells

(Fig. 1e) and observed the expression of tdTom after PHx in the liver. Consistent with the immunofluorescence data, the expression of tdTom significantly increased in HNF4α-expressing hepatocytes but not in CK19-expressing ductal cells after PHx (Fig. 1f). Indeed, ~41% of HNF4a-expressing hepatocytes expressed tdTom (Fig. 1g). These data demonstrate that the expression of PROM1 significantly increases in hepatocytes during liver regeneration after PHx.

### PROM1 deficiency impairs liver regeneration in mice

To determine the role of PROM1 in the process of liver regeneration, we compared the livers of wild-type (*Prom1^f/f*) and liver-specific *Prom1* knockout mice (*Prom1^LKO*) after PHx. As a result of measuring the remnant liver-to-body weight ratio following PHx, liver regeneration of *Prom1^LKO* mice was impaired compared to that of *Prom1^f/f* mice (Fig. 2a). *Prom1^f/f* mice recovered their original liver mass almost 5 days after PHx, whereas *Prom1^LKO* mice did not. Compared with *Prom1^f/f* mice, the liver-to-body weight ratio was significantly lower in *Prom1^LKO* mice 48 and 120 hours after surgery.

To investigate hepatocyte proliferation between *Prom1^f/f* and *Prom1^LKO* mice during liver regeneration, we confirmed cell cycle-related genes (Cyclin A, B, D, E, and PCNA) in PHx livers by qRT-PCR and immunoblotting. The levels of cyclin mRNAs were reduced in *Prom1^LKO* livers more than in *Prom1^f/f* livers (Fig. 2b). Consistently, the expression of cell cycle-related proteins in *Prom1^LKO* mice decreased compared to that in *Prom1^f/f* mice after PHx (Fig. 2c, d). We also analyzed hepatocyte proliferation by H&E staining and double immunofluorescence along with Ki-67 (as a cell proliferation marker) and HNF4α (Fig. 2e–g). As shown in Fig. 2f, Ki-67 expression in *Prom1^f/f* livers increased more than that in *Prom1^LKO* livers after PHx. Indeed, PROM1 deficiency reduced the number of Ki-67-positive cells by ~50% (Fig. 2g). These results suggested that the liver-specific deletion of PROM1 decreased hepatocyte proliferation and impaired liver regeneration after PHx.

### Liver-specific PROM1 deficiency reduces liver regeneration in mice injected with $CCl_4$

To further investigate the effects of PROM1 deficiency on the proliferation of hepatocytes in the regenerating liver, we analyzed the liver after injecting $CCl_4$ into mice. As with liver regeneration by PHx, PROM1 expression also increased after $CCl_4$ injection. PROM1 mRNA increased over ~10-fold in the liver by $CCl_4$ (Fig. 3a). PROM1 double immunofluorescence with HNF4α or CK19 showed that major cells expressing PROM1 were hepatocytes after $CCl_4$ injection (Fig. 3b, c).

Next, we compared the expression of cell cycle-related proteins in the livers of *Prom1^f/f* and *Prom1^LKO* mice after $CCl_4$ injection. PROM1 deficiency significantly decreased the expression of Cyclin A, Cyclin B, and PCNA, as determined by immunoblotting (Fig. 3d, e). Hepatocyte proliferation and apoptosis were confirmed by H&E staining and TUNNEL assay in the livers of *Prom1^f/f* and *Prom1^LKO* mice after $CCl_4$ injection (Fig. 3f and Supplementary Fig. 2). PROM1 deficiency decreased the number of Ki-67-expressing cells without changing apoptosis after $CCl_4$ injection by ~80%, as determined by immunofluorescence (Fig. 3g, h and Supplementary Fig. 2). Taken together, these data suggested that PROM1 deficiency attenuates hepatocyte proliferation during liver regeneration in the $CCl_4$ model.

### PROM1 increases IL-6 signaling during liver regeneration

Hepatocyte proliferation in the early stage of liver regeneration requires the JAK-STAT, PI3K, MAPK, and β-catenin signaling pathways initiated by different mitogens, such as IL-6, EGF, HGF and Wnt[3,28,29]. To examine the signaling pathways affected by PROM1, we observed the expression and activation of these mitogenic signaling molecules after PHx by immunoblotting. PROM1 deficiency significantly decreased the phosphorylation status of STAT3 and ERK but not the phosphorylation status of AKT or GSK3β (Fig. 4a, b). In the $CCl_4$ model, PROM1

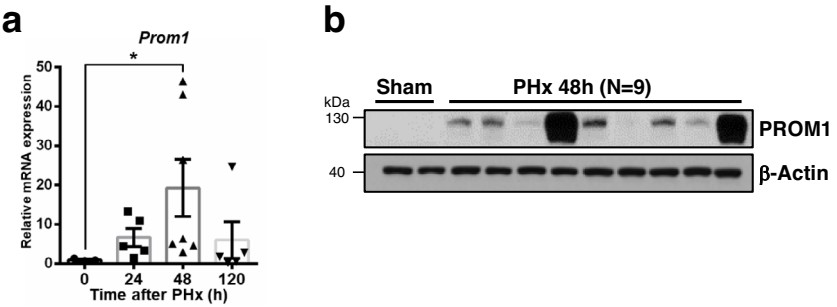

**Fig. 1 | The expression of PROM1 in hepatocytes increases after partial hepatectomy.** A 2/3 partial hepatectomy was performed in 8-week-old male wild-type mice. **a** The relative mRNA level of PROM1 in sham and PHx livers ($n = 3$ for sham, $n = 5$ for 24 h, $n = 7$ for 48 h, $n = 5$ for 120 h, $p = 0.045$). **b** Immunoblotting for PROM1 in wild-type livers 48 hours after PHx ($n = 2$ for sham, $n = 9$ for PHx). **c**, **d** Double immunofluorescence for PROM1 and HNF4α (**c**) or CK19 (**d**) in sham and PHx livers. **e** $Prom1^{Cre/ERT2};Rosa26^{tdTomato}$ mice were generated for lineage tracing of cells expressing PROM1 in the liver. PHx was performed 1 day after tamoxifen injection. The mice were analyzed 7 days after sham ($n = 4$) or PHx ($n = 4$). **f** Representative images of tdTom double immunofluorescence with HNF4α or CK19 in sham and PHx livers. **g** The percentage of tdTom-expressing cells was statistically determined from total HNF4α- or CK19- expressing cells ($p = 6.197 \times 10^{-6}$ for HNF4α). Scale bar = 100 μm. Two-sided student $t$-test; *$p < 0.05$, ***$p < 0.001$, n.s, nonsignificant. Data are expressed as the mean ± SEM with individual values. Source data are provided as a Source Data file.

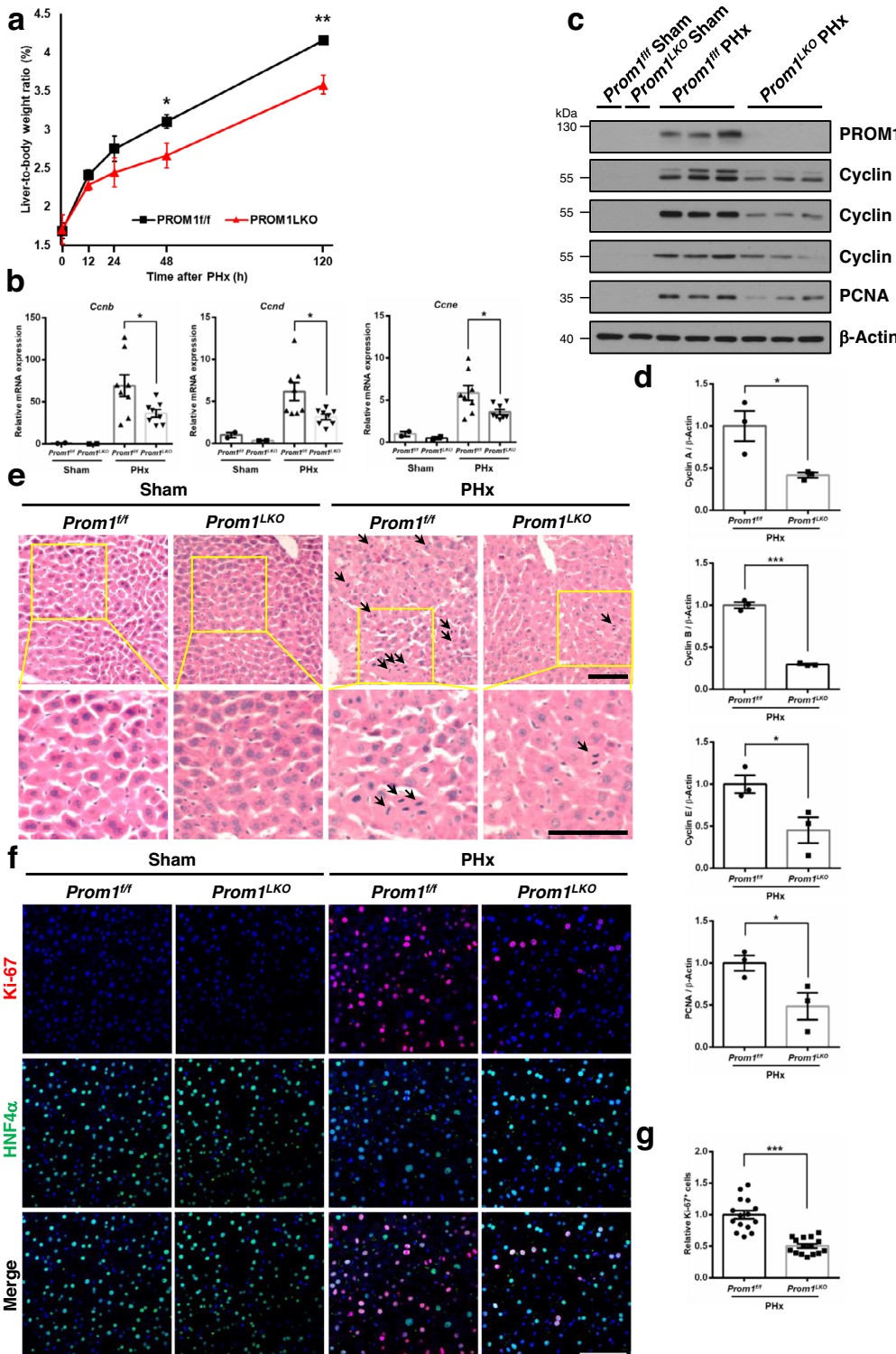

**Fig. 2 | Liver-specific deletion of Prom1 in mice impairs liver regeneration after partial hepatectomy.** A 2/3 partial hepatectomy was performed in 8-week-old male *Prom1^f/f* and *Prom1^LKO* mice. **a** Ratio of liver-to-body weight on the indicated days after PHx (*n* = 4 for sham, 12 h, and 120 h, *n* = 5 for 24 h, *n* = 7 for 48 h, *p* = 0.034 for 48 h, *p* = 0.005 for 120 h). **b** The relative mRNA levels of cell cycle genes (*Ccnd, Ccne, Ccnb*) 48 hours after PHx (*n* = 8). Each mRNA level was normalized by 18 S rRNA (*p* = 0.041 for *Ccnb*, *p* = 0.029 for *Ccnd*, *p* = 0.039 for *Ccne*). **c** Immunoblotting for PROM1 and cell cycle proteins (Cyclin A, B, and E, and PCNA) 48 hours after PHx. **d** Statistical analysis of the band intensity in **c** (*n* = 3 independent samples, *p* = 0.032 for Cyclin A, *p* = 4.508 × 10⁻⁵ for Cyclin B, *p* = 0.042 for

Cyclin E, *p* = 0.049 for PCNA). The band intensity of each protein was normalized to that of β-actin. **e** Representative H&E staining in the liver 48 hours after PHx. Mitotic cells are indicated by arrows. The experiment was repeated independently three times with similar results. **f** Representative double immunofluorescence for Ki-67 and HNF4α in the liver 48 hours after PHx. **g** Statistical analysis of the number of Ki-67-expressing cells after PHx (*n* = 3, *p* = 1.149 × 10⁻⁶). The number of Ki-67-positive cells was normalized to the number of DAPI-stained dots. Scale bar = 100 μm. Two-sided student *t*-test; **p* < 0.05, ***p* < 0.01, ****p* < 0.001. Data are expressed as the mean ± SEM with individual values. Source data are provided as a Source Data file.

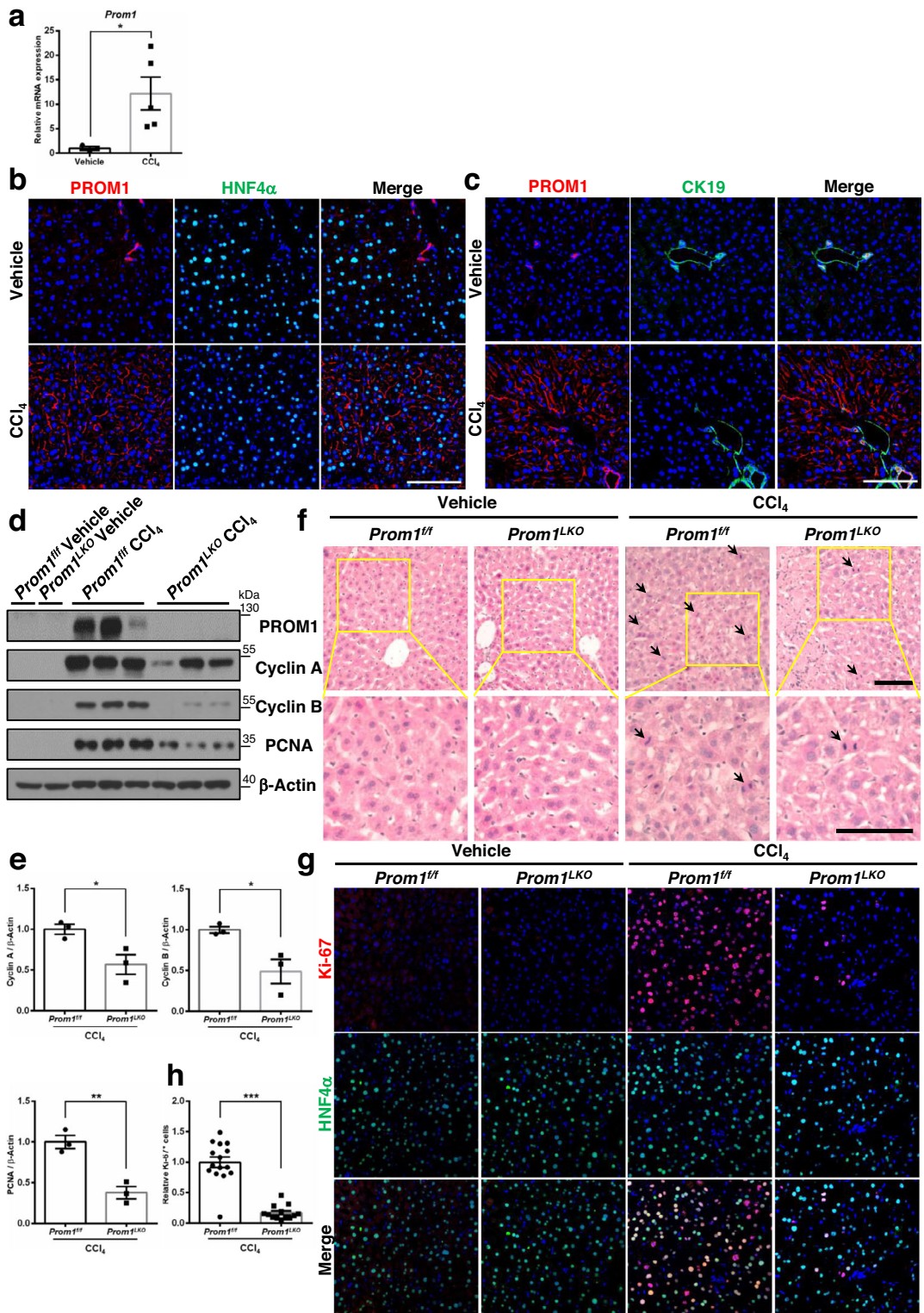

**Fig. 3 | Liver-specific deletion of Prom1 in mice impairs liver regeneration after CCl₄ injection.** Eight-week-old male *Prom1^f/f* and *Prom1^LKO* mice were intraperitoneally injected with vehicle (*n* = 3) or CCl₄ (*n* = 5) for 48 hours. The liver was analyzed by qRT-PCR, immunoblotting and immunofluorescence. **a** The relative mRNA level for PROM1. The mRNA level of PROM1 was normalized by 18 S rRNA (*p* = 0.028). **b, c** Double immunofluorescence for PROM1 and HNF4α (**b**) or CK19 (**c**). The experiment was repeated independently three times with similar results. **d** Immunoblotting for PROM1, Cyclin A and B, and PCNA. **e** Statistical analysis of the band intensity in **d** (*n* = 3, *p* = 0.033 for Cyclin A, *p* = 0.029 for Cyclin B,

*p* = 0.005 for PCNA). The band intensity of each protein was normalized to that of β-actin. **f** Representative H&E staining in the liver. Mitotic cells are indicated by arrows. The experiment was repeated independently three times with similar results. **g** Double immunofluorescence for Ki-67 and HNF4α. **h** Statistical analysis of the number of Ki-67-expressing cells (*n* = 3, *p* = 4.821 × 10⁻⁸). The number of Ki-67-positive cells was normalized to the number of DAPI-stained dots. Scale bar = 100 μm. Two-sided student *t*-test; **p* < 0.05, ***p* < 0.01, ****p* < 0.001. Data are expressed as the mean ± SEM with individual values. Source data are provided as a Source Data file.

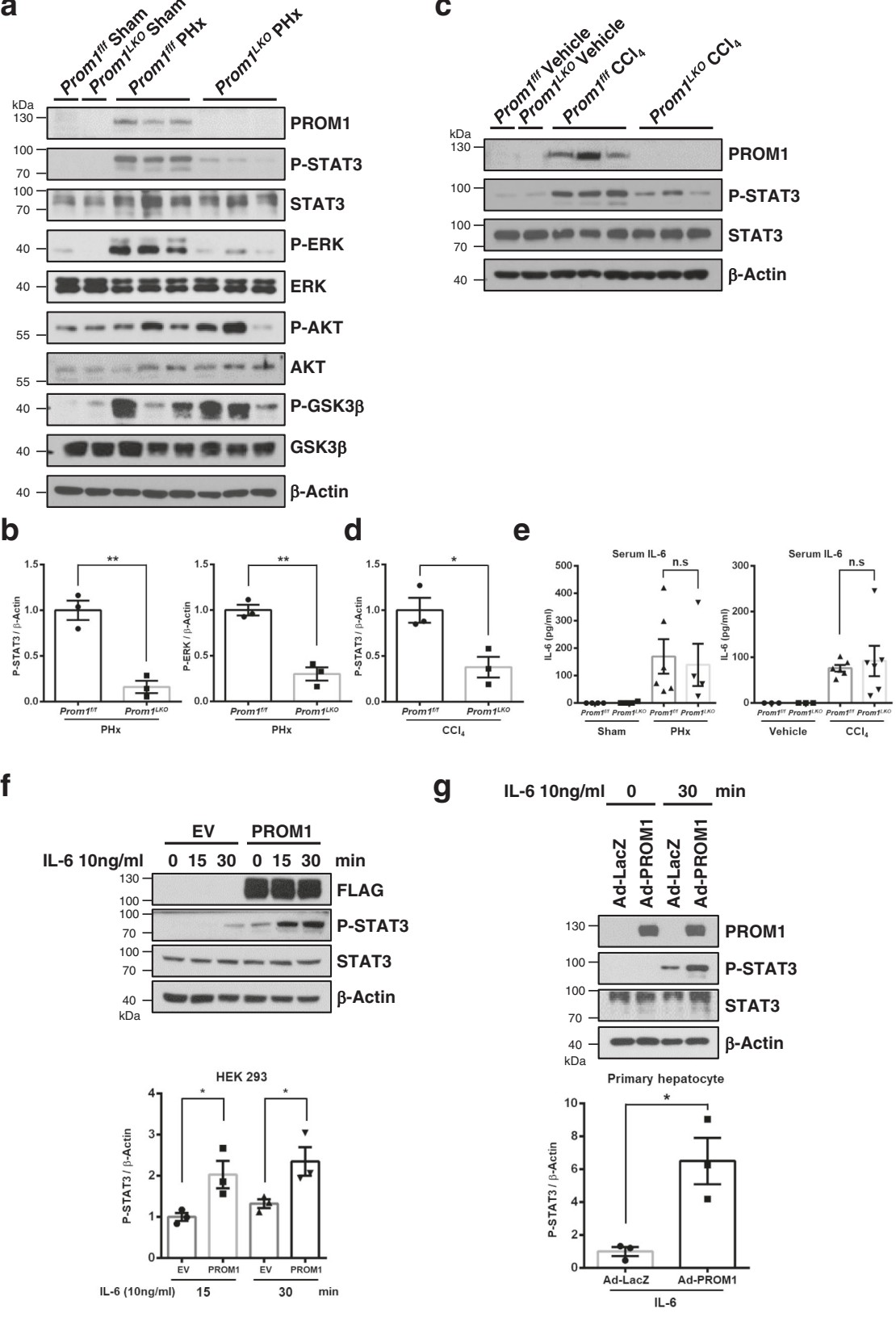

deficiency also decreased the phosphorylation of STAT3 (Fig. 4c, d). Since IL-6 signals are known to activate both STAT3 and ERK, these results led us to investigate the IL-6 signaling pathway in more detail. IL-6 ELISA showed that PROM1 deficiency did not change the serum level of IL-6 after PHx or CCl₄ injection (Fig. 4e). In addition, PROM1 deficiency did not change the expression level of growth factors such

as EGF and HGF as determined by qRT-PCR (Supplementary Fig. 3A, B). Thus, these data allow us to rule out the effect of PROM1 on IL-6, EGF and HGF production and secretion during liver regeneration. Next, we confirmed that PROM1 overexpression statistically increased IL-6-induced phosphorylation of STAT3 by ~2-fold in HEK 293 cells and by ~6-fold in primary hepatocytes obtained from $Prom1^{LKO}$ mice (Fig. 4f, g).

**Fig. 4 | PROM1 facilitates IL-6-STAT3 signaling pathway. a, b** A 2/3 partial hepatectomy was performed in 8-week-old male *Prom1^f/f* and *Prom1^LKO* mice. The liver was analyzed by immunoblotting 24 hours after PHx. Immunoblotting for PROM1, STAT3, P-STAT3, ERK, P-ERK, AKT, P-AKT, GSK3β, and P-GSK3β (**a**). Statistical analysis of the band intensity of P-STAT3 and P-ERK. The band intensity of each protein was normalized to that of β-actin (*n* = 3 independent samples, *p* = 0.003 for P-STAT3, *p* = 0.002 for P-ERK) (**b**). **c, d** Eight-week-old male *Prom1^f/f* and *Prom1^LKO* mice were intraperitoneally injected with vehicle (*n* = 3) or CCl₄ (*n* = 5) for 48 hours. Immunoblotting for PROM1, STAT3, and P-STAT3 (**c**). Statistical analysis of the band intensity of P-STAT3. The band intensity of each protein was normalized to that of β-actin (*n* = 3 independent samples, *p* = 0.025) (**d**). **e** Quantification of serum IL-6 in *Prom1^f/f* and *Prom1^LKO* mice 24 hours after PHx (Left) and 48 hours after CCl₄ injection (Right). *n* = 6 for *Prom1^f/f* PHx, CCl₄ mice and

*Prom1^LKO* CCl₄ mice, *n* = 4 for *Prom1^LKO* PHx mice **f, g** Empty vector (EV) or FLAG-tagged PROM1 was transfected into HEK293 cells for 48 hours. After serum starvation for 16 hours, cells were treated with 10 ng/ml human recombinant IL-6 for 0, 15, and 30 minutes (**f**). Primary hepatocytes were isolated from 8-week-old male *Prom1^LKO* mice. The cells were infected with adeno-LacZ or PROM1 for 16 hours, followed by serum starvation for 16 hours and then harvested after treatment with 10 ng/ml IL-6 for 30 minutes (**g**). Each experiment was independently repeated three times. Immunoblotting for FLAG or PROM1, STAT3 and P-STAT3. Statistical analysis of the band intensity of P-STAT3. The intensity of P-STAT3 was normalized to that of β-actin. *n* = 3 independent experiments (*p* = 0.041 for HEK 293 15 min, *p* = 0.048 for HEK 293 30 min, *p* = 0.018 for primary hepatocytes). Two-sided student *t*-test; *\*p* < 0.05, *\*\*p* < 0.01, n.s, nonsignificant. Data are expressed as the mean ± SEM with individual values. Source data are provided as a Source Data file.

To further confirm the association between PROM1 and the IL-6 signaling pathway, we observed whether liver regeneration impaired by PROM1 deficiency was rescued through adenoviral overexpression of constitutively activated STAT3 (Stat3c) in *Prom1^LKO* mice. As determined by qRT-PCR and immunoblotting (Supplementary Fig. 4A–C), cyclins A, B, and E and PCNA were significantly increased by Stat3c overexpression. Consistent with these data, Ki-67 and HNF4α immunofluorescence showed that hepatocyte proliferation was increased by Stat3c in *Prom1^LKO* mice because the number of Ki-67-expressing cells increased by ~4-fold (Supplementary Fig. 4D, E). These results suggested that PROM1 regulates hepatocyte proliferation through the IL-6 signaling pathway during liver regeneration.

## PROM1 regulates IL-6 signaling by interacting with GP130 in lipid rafts

GP130, a common receptor of the IL-6 receptor family and known as the IL-6 receptor beta-subunit signal transducer, associates with downstream molecules in lipid rafts for efficient signaling[30,31]. Since both PROM1 and IL-6 signaling complexes were in lipid rafts, PROM1 would bind to GP130. To examine the possibility, we investigated whether raft localization of GP130 is dependent on PROM1 in sham and PHx livers (Fig. 5a, b). PROM1 was expressed in lipid rafts of *Prom1^f/f* sham livers. However, GP130 was not expressed in lipid rafts of both *Prom1^f/f* and *Prom1^LKO* sham livers. In contrast to sham livers, both PROM1 and GP130 were found in lipid rafts of *Prom1^f/f* PHx livers. We confirmed that PROM1 deficiency reduced the expression of GP130 in lipid rafts of PHx livers. PROM1-positive hepatocytes accounted for ~1% of total hepatocytes in sham liver and ~40% in PHx liver (Fig. 1f, g). Thus, PROM1 expression level is too low to recruit GP130 to lipid rafts in sham liver. However, the upregulated PROM1 recruited GP130 to lipid rafts in PHx liver. Indeed, PROM1 overexpression in *Prom1^LKO* sham livers increased the expression of GP130 in liver lipid rafts (Fig. 5c).

Because giant plasma membrane vesicles (GPMVs), which are isolated from mammalian cells without using detergents, are useful for identifying raft proteins[32,33], we tested whether PROM1 and GP130 are recruited into raft phase of GPMV. HEK 293 cells were overexpressed with PROM1-GFP or glycosylphosphatidylinositol-anchored GFP (GPI-GFP) and stained with DiI (1,1'-Dioctadecyl-3,3,3',3'-tetramethylindocarbocyanine perchlorate) for a non-raft marker. As shown in Supplementary Fig. 5A, GPMVs were successfully isolated and partitioned to two phases because GPI-GFP and DiI signals were not co-localized with each other. Unexpectedly, PROM1-GFP was co-localized with DiI in the same phase. Because wheat germ agglutinin (WGA) is known to bind glycoproteins[34] and co-localize with PROM1 microdomain[15], we tried to investigate whether WGA-labeled membranes were located at the raft phase in GPMV. WGA existed in nonraft phase with PROM1-tdTom in Supplementary Fig. 5A. Our results suggest that PROM1 and WGA-labeled proteins were located to nonraft phase in GPMV. Although WGA-binding glycoproteins, including PROM1 are detergent-resistant membrane protein and have a typical

punctate staining pattern for lipid raft proteins[15], they exist at nonraft phase in GPMV.

Next, we demonstrated the molecular interaction between PROM1 and GP130 by immunoprecipitation. As shown in Fig. 5d–f, endogenous immunoprecipitation in PHx wild-type liver and reciprocal exogenous immunoprecipitation in HEK 293 cells showed a molecular interaction between PROM1 and GP130. Next, we investigated the interaction of PROM1 and GP130 after membrane cholesterol depletion. After methyl-beta-cyclodextrin (MβCD) treatment, whole cell lysates were obtained using NP-40 or Brij-35 from HEK 293 cells overexpressing PROM1-FLAG and GP130-His. Cholesterol depletion was confirmed by PROM1 and GP130 solubility in Brij-35 (Supplementary Fig. 5B). Coimmunoprecipitation experiments using NP-40 lysates showed that the interaction between PROM1 and GP130 was not changed by MβCD treatment (Supplementary Fig. 5B). These data suggested that the interaction between PROM1 and GP130 was not involved in lipid rafts.

Next, we determined cellular localization of GP130 in the presence of PROM1. GP130-His was overexpressed in HEK293 cells along with or without PROM1-FLAG. The localization of GP130 and PROM1 was determined by immunofluorescence after labeling wheat germ agglutinin (WGA), which is co-localized with PROM1[15]. As shown in Fig. 5g, in the absence of PROM1, GP130 was mainly found in intracellular compartments, which was not labeled with WGA. In the presence of PROM1, GP130 was co-localized with PROM1 in the plasma membrane, which was labeled with WGA. All these data indicate that PROM1 and GP130 forms a complex, which is important for the raft localization of GP130.

## The first extracellular domain of PROM1 is required for the interaction with GP130 and the regulation of the STAT3 signaling pathway

To determine the domain required for the interaction between PROM1 and GP130, we generated various deletion mutants of PROM1 (Fig. 6a). A co-immunoprecipitation assay using these mutants showed that all deletion mutants of PROM1 still interacted with GP130 (Fig. 6b). Based on this result, the first extracellular domain of PROM1 (PROM1-EX1) would be an important region for the interaction between the two proteins. To examine the possibility, we generated GPI-anchored PROM1-EX1 (PROM1^GPI-EX1) in which the first transmembrane domain was substituted with a GPI anchor and observed the interaction between PROM1^GPI-EX1 and GP130. We identified the GPI anchor from PROM1^GPI-EX1 after treatment of phosphatidylinositol-specific phospholipase C (PI-PLC) which releases GPI-anchored proteins from plasma membrane (Supplementary Fig. 6A, B). As shown Fig. 6c, EX1 itself interacted with GP130.

PROM1 is not co-localized with alkaline phosphatase, a GPI-anchored protein, although both proteins are raft proteins[15], suggesting that there are two types of rafts, PROM1-enriched and GPI-anchored protein-enriched rafts. To further address this question, cell-surface immunostaining for PROM1 and PROM1^GPI-EX1 was performed in HEK 293

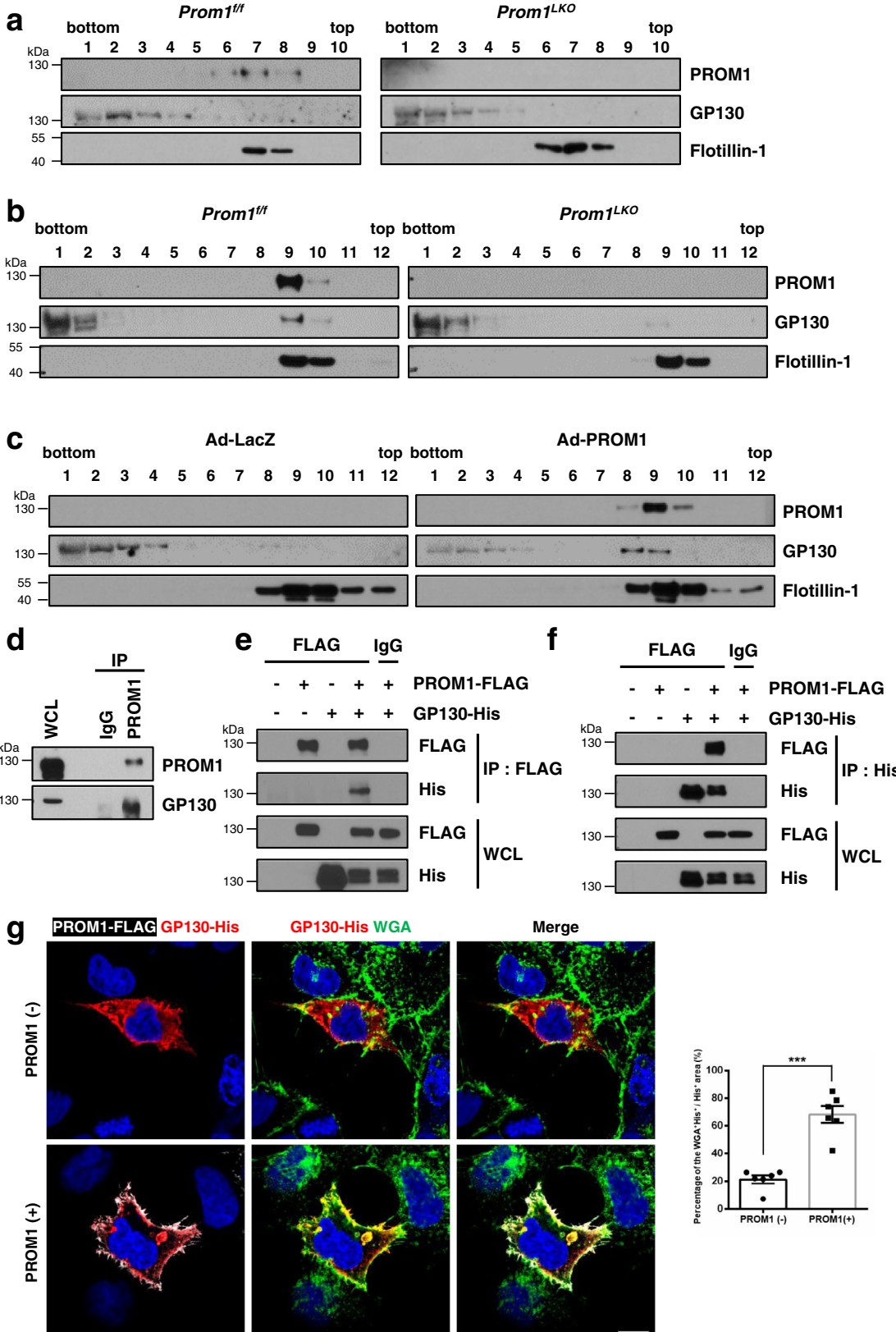

cells overexpressing either or both of untagged PROM1 and FLAG-tagged PROM1^GPI-EX1. PROM1 was labeled with an anti-PROM1 antibody that binds to EX2 and EX3 whereas PROM1^GPI-EX1 was labeled with an anti-FLAG antibody. As shown in Supplementary Fig. 6C, PROM1 and PROM1^GPI-EX1 were not co-localized with each other. In addition, WGA was co-localized with PROM1 but not with PROM1^GPI-EX1 (Supplementary

Fig. 6D). All these data suggested that PROM1 and PROM1^GPI-EX1 were not co-localized in the same type of lipid rafts.

Since GPI-anchored proteins are expressed in lipid rafts, we examined whether PROM1^GPI-EX1 enhances the STAT3 signaling pathway. Exogenous PROM1^GPI-EX1 itself increased the activity of STAT3 in HEK 293 cells (Fig. 6d, e). Taken together, the first extracellular domain

**Fig. 5 | PROM1 interacts with GP130 in lipid rafts. a, b** Detergent-resistant lipid rafts were isolated from *Prom1^f/f* and *Prom1^LKO* male mouse livers 48 hours after sham (**a**) and PHx (**b**). Protein expression levels of PROM1, GP130, and Flotillin-1 were determined by immunoblotting in each fraction after sucrose gradient ultracentrifugation. **c** Detergent-resistant lipid rafts were isolated from 8-week-old male *Prom1^LKO* mice 3 days after infection with adeno-LacZ or adeno-PROM1. Immunoblotting for PROM1, GP130, and Flotillin-1 in each fraction. **d** Co-immunoprecipitation was performed with normal IgG or anti-PROM1 in wild-type livers 48 h after PHx. Immunoblotting for endogenous PROM1 and GP130. **e, f** The molecular interaction between PROM1 and GP130 was determined by reciprocal immunoprecipitation after PROM1-FLAG and GP130-His were transfected into HEK

293 cells for 48 hours. **g** GP130-His was overexpressed in HEK 293 cells along with or without PROM1-FLAG. The localization of GP130 and PROM1 was determined by immunofluorescence for His and FLAG after WGA labeling (Left). The percentages of the WGA and His double positive area were analyzed statistically in 6 images per group. $n = 6$ independent cells, $p = 4.268 \times 10^{-5}$ (Right). The percentages of the WGA and His double positive area were normalized to total His positive signals. Scale bar = 10 μm. Two-sided student *t*-test; ***$p < 0.001$. Data are expressed as the mean ± SEM with individual values. Source data are provided as a Source Data file. WCL, whole cell lysates; IP, immunoprecipitation; IgG, normal IgG; WGA, Wheat germ agglutinin.

of PROM1 is required for binding to GP130 and regulating the GP130-STAT3 signaling pathway.

**The expression of GPI-anchored PROM1-EX1 rescues liver regeneration in PROM1-deficient mice after partial hepatectomy**
To evaluate whether PROM1^GPI-EX1 has an in vivo function in liver regeneration after PHx, we observed recovery of liver mass and hepatocyte proliferation after adenoviral overexpression of PROM1^GPI-EX1 in *Prom1^LKO* mice. PROM1^GPI-EX1 was overexpressed in the liver, as determined by qRT-PCR and immunoblotting (Fig. 7b, c). The overexpression of PROM1^GPI-EX1 alone was sufficient to increase the liver-to-body weight ratio at 24 and 48 hours after PHx in *Prom1^LKO* mice (Fig. 7a). As determined by qRT-PCR and/or immunoblotting for Cyclin A and B, PCNA and phospho-STAT3, H&E staining, and immunofluorescence for Ki-67, the overexpression of PROM1^GPI-EX1 statistically increased hepatocyte proliferation via STAT3 phosphorylation compared to the overexpression of LacZ (Fig. 7b–h and Supplementary Fig. 7). In addition, GP130 was relocalized into lipid rafts after the overexpression of PROM1^GPI-EX1 (Fig. 7i). These data suggested that PROM1^GPI-EX1 has a crucial role in refining GP130 into lipid rafts and mediating an IL-6-GP130 axis, thereby promoting liver regeneration.

## Discussion
PROM1 is well known as a marker for cancer stem cells and normal stem cells. Recent studies have revealed its ability to regulate various cellular signal transduction pathways by interacting with PI3K, HDAC6, radixin, and SMAD7[26,27,35,36]. Indeed, PROM1-deficiency leads to the prevention of glucagon-induced gluconeogenesis via inactivating the function of radixin as A kinase-anchoring protein (AKAP)[26], and aggravation of bile duct ligation (BDL)-induced liver fibrosis via SMAD7 degradation[27], indicating that PROM1 has different functions in the liver. Here, we demonstrated that PROM1 is also necessary for regulating IL-6 signaling during liver regeneration. We found that the expression of PROM1 dramatically increased in hepatocytes during liver regeneration after PHx or CCl₄ injection. Hepatocellular PROM1 facilitated the IL-6 signaling pathway by interacting with GP130 in lipid rafts. As a result, PROM1 promoted the proliferation of hepatocytes during liver regeneration (Fig. 8). Thus, this study is the first to elucidate the function of PROM1 in liver regeneration and is expected to provide a deeper understanding of liver regeneration and liver transplantation therapy.

Lipid rafts are defined as a membrane domain resistant to non-ionic detergent because they are tightly packed with glycosphingolipids and cholesterol[37–39]. The tight packaging of glycosphingolipids and cholesterol also induces phase partitioning in GPMV[33,40]. However, phase partitioning in GPMV is different from DRM for many multi-span transmembrane proteins due to the disruption of the cytoskeleton network, lipid bilayer asymmetry, and protein-protein interaction in GPMV[41]. In previous our report[26], PROM1 interacts with cortical actin by radixin. Since cortical actin might be disrupted during GPMV isolation, it seems that we failed to observe raft recruitment of PROM1 in GPMV.

GP130 is recruited from nonraft to lipid rafts after ciliary neurotrophic factor (CNTF) treatment in neural cells (IMR-32 cells)[42] whereas it is always found in lipid rafts independent on ligand activation in mouse embryonic neural precursor cells[43], Madin-Darby canine kidney cells[44], and Hep3B cells[30]. Our data showed that GP130 was found in lipid rafts in PROM1-overexpressing *Prom1^LKO* sham liver, indicating GP130 localization in lipid rafts is dependent on PROM1 but not IL-6 activation.

During liver regeneration after PHx and CCl₄ injection, PROM1 was highly upregulated, as determined by qRT-PCR, immunofluorescence, and immunoblotting. In addition, PROM1 upregulation was dramatically demonstrated in PROM1 lineage tracing mice (*Prom1^Cre/ERT2*; *Rosa26^tdTomato* mice), in which cells express tdTom under the control of the PROM1 promoter. Because hepatocellular PROM1 upregulation is also observed after bile duct ligation (BDL)[27] and a lithogenic diet (data not shown), various liver damages might lead to hepatocellular PROM1 upregulation. Many extracellular and intracellular factors, such as HIF-1α, TGFβ1, p53 and mTOR, regulate the expression of PROM1[21,45]. A previous study reported that STAT3 promotes the transcription of PROM1 in hepatocellular carcinoma[46]. Therefore, we hypothesized that the IL-6-STAT3 signaling pathway might be necessary for upregulating PROM1 in hepatocytes, and then, the upregulated PROM1 would form a 'positive loop' because PROM1 promotes the IL-6-STAT3 signaling pathway.

PROM1 interacts with various signaling molecules through its different domains. The cytoplasmic C-terminal domain of PROM1 binds to PI3K and radixin, maintaining cancer stem cell properties and regulating glucagon-induced PKA activity, respectively[26,35]. The first intracellular loop of PROM1 binds to HDAC6 and SMAD7, regulating β-Catenin signaling and TGFβ signaling, respectively[27,36]. Here, we demonstrated that the first extracellular domain of PROM1 binds to GP130. Furthermore, lipid raft-targeted PROM1^GPI-EX1 alone is sufficient to replace the function of full-length PROM1, which recruits GP130 into lipid rafts and then facilitates IL-6-induced STAT3 phosphorylation, leading to hepatocyte proliferation and liver regeneration.

The PROM1-positive population in various tumors has self-renewal and differentiation potential and chemotherapy or radiotherapy resistance[17–20]. Although most cancers are removed through cancer therapy, only a small number of surviving PROM1-positive cells can proliferate and cause cancer to recur. Thus, PROM1 has been considered a very important target protein for cancer therapy. Because PROM1 expression was upregulated at the early stage of liver regeneration (within 48 hours after PHx) and then returned to sham liver levels at the termination stage of regeneration (7 days after PHx, data not shown), hyperplasia or tumorigenesis might not occur during liver regeneration.

In addition to IL-6, GP130 is involved in various signaling pathways of IL-6 family cytokines, such as IL-11, leukemia inhibitory factor (LIF), oncostatin M, and ciliary neurotrophic factor[47]. Therefore, the PROM1-GP130 axis could be a potential therapeutic target for human diseases induced by these cytokines. For example, a PROM1-neutralizing antibody targeting PROM1-EX1 is a good candidate for alleviating inflammatory diseases caused by these cytokines.

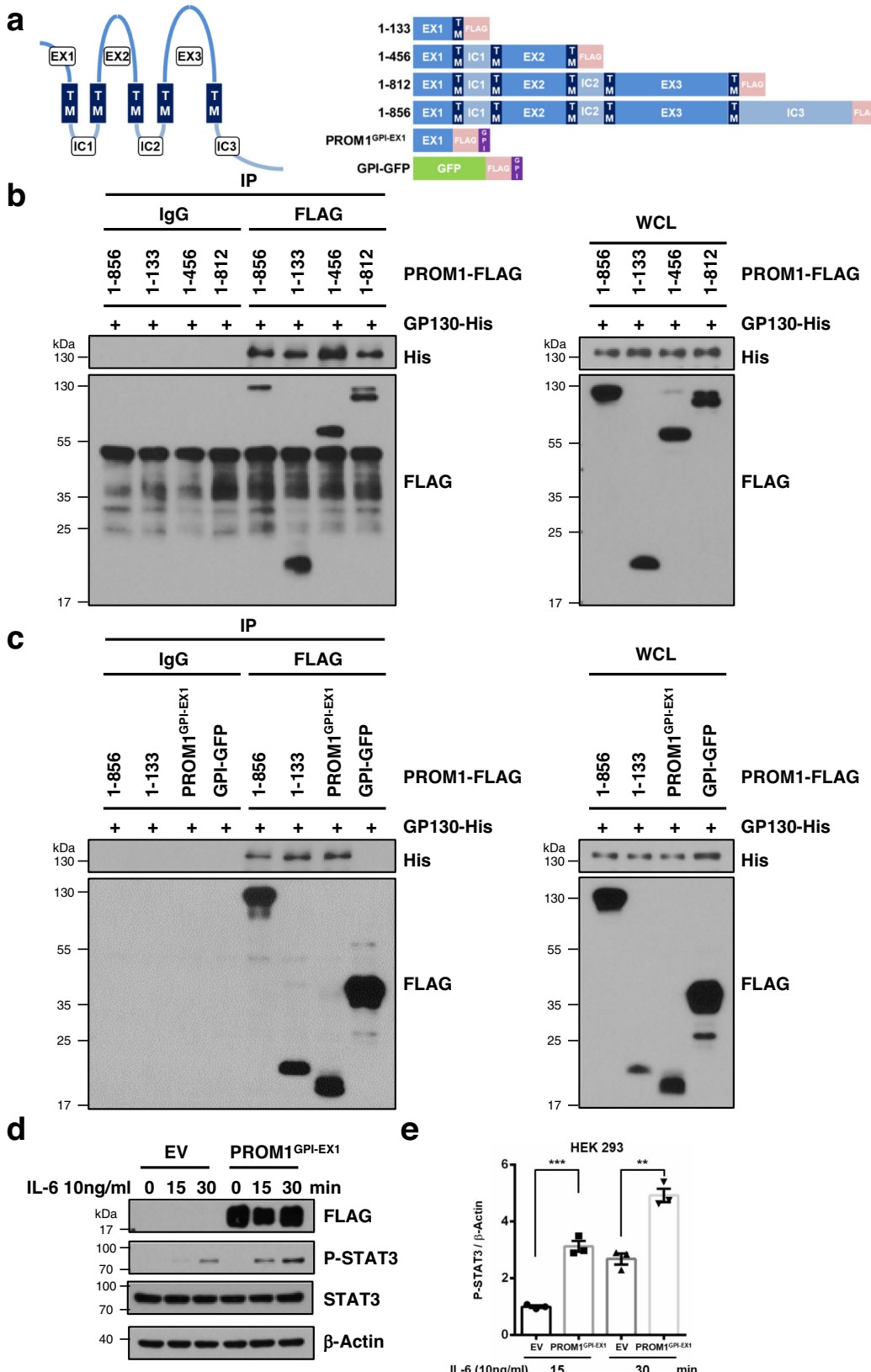

## Methods

### Animal studies

Whole-body *Prom1* knockout mice (*Prom1^{Cre/ERT2-nlacZ}*) were purchased from The Jackson Laboratory (Stock No: 017743, Bar Harbor, ME, USA) and backcrossed with C57BL/6 N mice for five generations. Liver-specific *Prom1* knockout mice were generated by crossing *Prom1^{flox/flox}* C57BL/6 mice (ToolGen, Seoul, Korea) with *Alb-Cre* C57BL/6 mice containing the Cre recombinase sequence driven by the albumin promoter (The Jackson Laboratory, Bar Harbor, ME, USA). *Prom1* lineage tracing mice were generated by crossing *Prom1^{Cre/ERT2-nlacZ}* C57BL/6 mice with *Rosa26^{tdTomato}* C57BL/6 mice containing the tdTomato sequence prevented by the loxP-flanked STOP cassette (Stock

**Fig. 6 | The first extracellular domain of PROM1 interacts with GP130 and regulates the STAT3 signaling pathway. a** Structures of PROM1 deletion mutants. EX extracellular domain, TM transmembrane domain, GPI glycosylpho-sphatidylinositol, GFP green fluorescence protein. **b, c** Co-immunoprecipitation between each PROM1 mutant and GP130. HEK 293 cells were transfected with various FLAG-tagged PROM1 mutants (1-133, 1-456, 1-812, and PROM1[GPI-EXI]) or full-length PROM1 (1-856) and His-tagged GP130 for 48 hours. Each experiment was repeated independently three times with similar results. **d, e** HEK 293 cells were transfected with empty vector (EV) or FLAG-tagged PROM1[GPI-EXI] for 48 hours. After serum starvation for 16 hours, HEK 293 cells were treated with 10 ng/ml IL-6 for 0, 15, or 30 minutes. The experiments were independently repeated three times. Immunoblotting for STAT3, P-STAT3 and FLAG (**d**). Statistical analysis of the band intensity of P-STAT3. The band intensity of P-STAT3 was normalized to that of β-actin. $n = 3$ independent experiments, $p = 3.970 \times 10^{-4}$ for 15 min, $p = 0.002$ for 30 min (**e**). WCL, whole cell lysates; IP, immunoprecipitation; IgG, normal IgG. Two-sided student $t$-test; **$p < 0.01$, ***$p < 0.001$. Data are expressed as the mean ± SEM with individual values. Source data are provided as a Source Data file.

No: 007914, The Jackson Laboratory, Bar Harbor, ME, USA). For Cre-loxP recombination, tamoxifen (T5648; Sigma, USA, 20 mg/ml in corn oil) was intraperitoneally injected at 150 mg/kg 1 day before 2/3 partial hepatectomy in 8-week-old male mice.

All mice were housed in plastic cages under a 12:12-hour light/dark photoperiod at controlled temperature with free access to water and food. All mice were bred, maintained, and cared for in a manner consistent with criteria outlined in the Principles of Laboratory Animal Care (NIH publication no. 85-23, revised 1985). Protocols for animal studies were approved by the Institutional Animal Care and Use Committee of Korea University and the Korean Animal Protection Law (KUIACUC-2019-0111).

To investigate liver regeneration in mice, a 2/3 partial hepatectomy and CCl$_4$ injection were performed. For the 2/3 partial hepatectomy model, two-thirds of the mouse liver was surgically removed as previously described[48,49]. Briefly, 8-week-old male mice were anesthetized using isoflurane. An abdominal midline incision was made to open the abdominal cavity to expose the liver. The hepatic left lateral and median lobes were isolated and ligated. After ligation, each lobe was removed with surgical scissors. Then, the abdominal skin was sutured and sterilized. For a sham operation, 8-week-old male mice were anesthetized with isoflurane and the liver was exposed by an abdominal midline incision. After liver exposure, the abdominal skin was sutured without lobe resection. After surgery, the mice were kept warm for recovery. For the CCl$_4$ model, 8-week-old male mice were intraperitoneally injected with 25% CCl$_4$ in corn oil (C8267; Sigma) at a dose of 2.4 μl/g body weight.

The gender of all mice used in each experiment was male.

### Mouse primary hepatocytes isolation

Primary hepatocyte isolation was performed based on two-step collagenase perfusion as previously described[26]. Briefly, 8-week-old male mice were anesthetized with avertin (T48402; sigma) intraperitoneal injection of 250 mg/kg body weight). After an abdominal midline incision, the livers were perfused with EGTA-containing perfusion buffer (140 mM NaCl, 6 mM KCl, 10 mM HEPES, and 0.08 mg/mL EGTA, pH 7.4) at a rate of 7 ml/min for 5 min, followed by continuous perfusion with collagenase-containing buffer (66.7 mM NaCl, 6.7 mM KCl, 5 mM HEPES, 0.48 mM CaCl$_2$, and 3 g/mL type IV collagenase, pH 7.4) for 8 min. After collecting parenchymal cells by low-speed (50 × $g$, 4 min) centrifugation, viable hepatocytes were purified by Percoll gradient centrifugation. Then, hepatocytes were resuspended in complete growth medium (M199 media containing 10% fetal bovine serum, 23 mM HEPES, and 10 nM dexamethasone) and seeded on collagen-coated plates at a density of $3.3 \times 10^5$ cells/ml. After 4 hours of cell attachment, the medium was replaced with complete growth medium and replaced daily before use in all experiments. For in vitro analysis of IL-6-induced STAT3 phosphorylation, cells were treated with human recombinant interleukin-6 (200-06; Peprotech, USA).

### Adenovirus preparation and infection

Adenoviruses harboring LacZ, PROM1, STAT3C (#99264, Addgene), and PROM1[GPI-EXI] were prepared as previously described[50]. AD293 cells were infected with each viral stock to amplify the viruses. Virus purification was performed by double cesium chloride-gradient ultracentrifugation. Viral particles in cesium chloride (density≒1.345) were collected and washed with washing buffer (10 mM Tris pH 8.0, 2 mM MgCl$_2$, and 5% sucrose). Purified adenoviruses ($0.5 \times 10^9$ pfu) were intravenously injected into the tails of mice.

### RNA isolation and quantitative RT−PCR

Total RNA was extracted from liver tissues using an easy-spin™ total RNA extraction kit (Intron Biotechnology, Korea) according to the manufacturer's protocol. Total RNA (4 μg) was used for cDNA synthesis using random hexamers, oligo dT primers, and reverse transcription master mix (EBT-1511, EBT-1512; ELPIS Biotech, Korea). Quantitative real-time PCR was performed using the cDNAs and each gene-specific oligonucleotide primer in the presence of TOPreal qPCR premix (RT500M; Enzynomics, Korea). The following real-time PCR conditions were used: an initial denaturation step at 95 °C for 15 min, followed by 50 cycles of denaturation at 95 °C for 10 sec, annealing at 58 °C for 15 sec, and elongation at 72 °C for 20 sec. Each PCR product was evaluated by melting curve analysis for quality control. The qRT-PCR data were collected using LightCycler 480 software 1.5.0 (Roche). Supplementary Table 1 shows the sequences of the gene-specific primers used for qRT-PCR.

### Immunoblotting and immunoprecipitation

To extract whole cell lysates, the livers were homogenized with a tissue homogenizer and harvested. The homogenized tissues were lysed with radioimmunoprecipitation assay buffer (50 mM Tris-Cl pH 8.0, 150 mM NaCl, 1% NP-40, 0.5% sodium deoxycholate, 0.1% sodium dodecyl sulfate, and protease- and phosphatase-inhibitor cocktail (P3100, P3200; Gendepot, USA)) on ice for 30 min. Whole-cell lysates were extracted from supernatant by microcentrifugation at 14,000 rpm for 10 min at 4 °C. The whole cell lysates were quantified by BCA assay. The normalized protein samples were separated by sodium dodecyl sulfate-polyacrylamide gel electrophoresis. The separated proteins were transferred to a nitrocellulose membrane and incubated with the primary antibodies of interest (Supplementary Table 2) followed by incubation with horseradish peroxidase (HRP)-conjugated secondary antibodies (Supplementary Table 3). The protein band signals were visualized by chemiluminescence detection using an EZ-Western kit (DG-W500; Dogenbio, Korea).

For immunoprecipitation, homogenized tissues or cells were lysed with buffer containing 25 mM HEPES, 150 mM NaCl, 1% NP-40, 10 mM MgCl$_2$, 1 mM EDTA, 2% glycerol, and protease inhibitor cocktail (P3100; Gendepot) on ice for 30 min. Whole-cell lysates were extracted from the supernatant by microcentrifugation at 14,000 rpm for 10 min at 4 °C. The whole cell lysates were quantified by BCA assay. One milligram of protein in whole cell lysates was incubated with specific primary antibody overnight at 4 °C, followed by incubation with 60 μg of Protein A- or G-agarose bead slurry (11134515001, 11243233001; Roche, Germany) for 4 hours at 4 °C. The bead precipitates were washed with buffer containing 25 mM HEPES, 150 mM NaCl, 1% NP-40, 10 mM MgCl$_2$, 1 mM EDTA, 2% glycerol, and protease inhibitor cocktail (P3100; Gendepot) 4 times. Protein samples were obtained from the precipitates and analyzed by immunoblotting as described above.

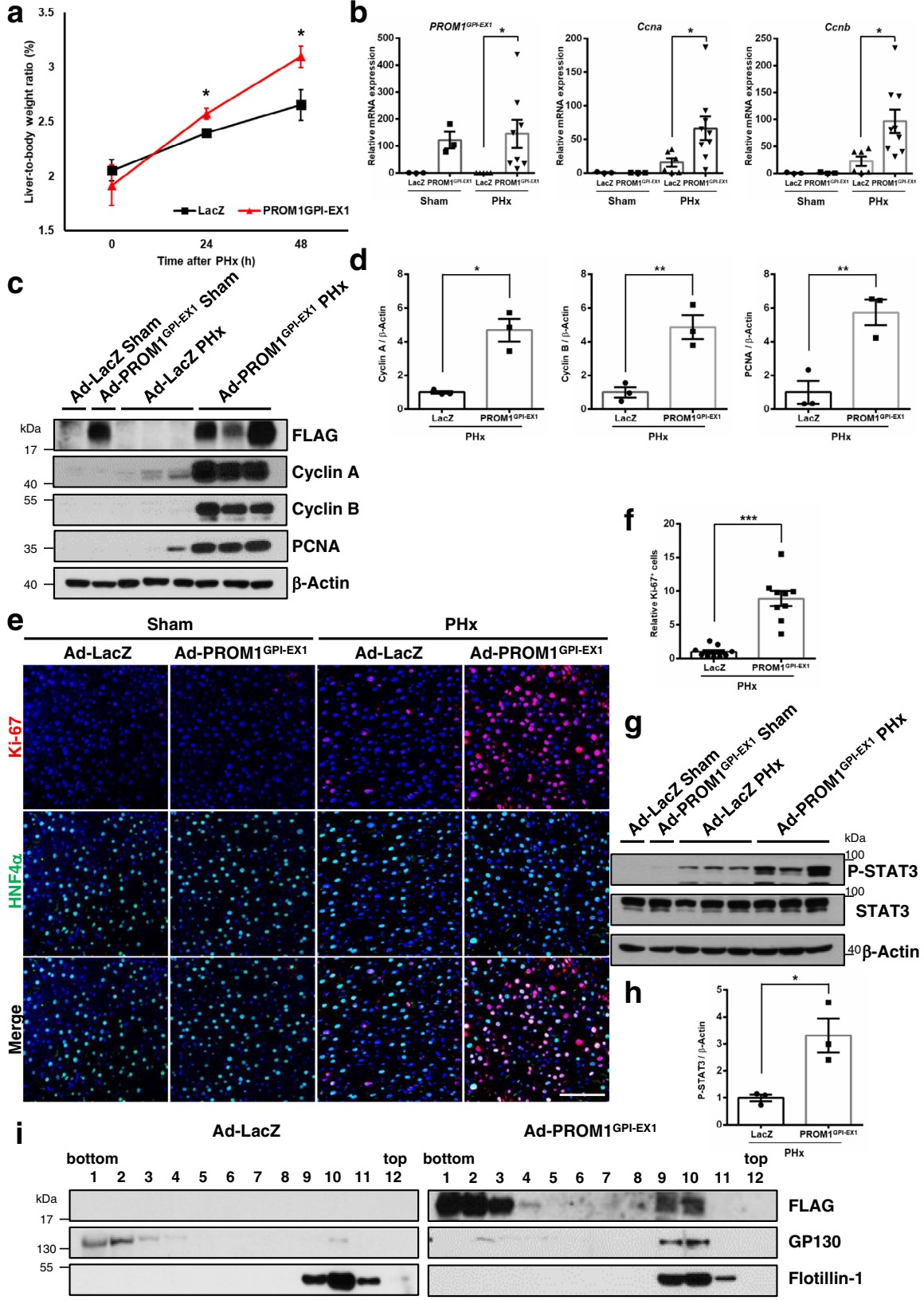

## Immunofluorescence staining

For immunofluorescence staining of liver tissues, frozen tissues were cut to a thickness of 5 μm using a cryocut microtome (Leica).

PROM1 immunofluorescence in the liver section using a rat monoclonal antibody (13A4) was performed as previously described[26]. In detail, for PROM1 double immunofluorescence with HNF4α or CK19,

the sections were incubated with proteinase K (0.06 U/mg) for 5 min, followed by blocking with 2.5% normal horse serum for 30 min at room temperature. Then, the sections were incubated with mouse anti-HNF4α (ab41898; Abcam, UK) or rabbit anti-CK19 (ab52625; Abcam) overnight at 4 °C. Next, for double immunofluorescence with PROM1, the sections were incubated with 4% paraformaldehyde in 0.1 M

**Fig. 7 | The expression of GPI-anchored PROM1-EX1 rescues liver regeneration in liver-specific Prom1-deficient mice.** A 2/3 partial hepatectomy was performed in 8-week-old male *Prom1^LKO* mice after infection with adeno-LacZ or adeno-PROM1^GPI-EX1–FLAG. **a** Ratio of liver-to-body weight on the indicated days after PHx. (*n* = 3 for sham, *n* = 6 for 24 h, *n* = 5 for 48 h, *p* = 0.021 for 24 h, *p* = 0.033 for 48 h) **b** The relative mRNA levels of *PROM1^GPI-EX1*, *Ccna*, and *Ccnb* in the liver 24 hours after PHx. Each mRNA level was normalized by 18 S rRNA. (*n* = 3 for sham, *n* = 6 for LacZ PHx, *n* = 9 for PROM1^GPI-EX1 PHx, *p* = 0.027 for *PROM1^GPI-EX1*, *p* = 0.022 for *Ccna*, *p* = 0.010 for *Ccnb*) **c, d** Immunoblotting for Cyclin A and B, PCNA, and FLAG in the liver 48 hours after PHx (**c**). Statistical analysis of the band intensities of Cyclin A and B and PCNA in **c**. The band intensity of each protein was normalized to that of β-actin (*n* = 3 independent samples, *p* = 0.029 for Cyclin A, *p* = 0.007 for Cyclin B, *p* = 0.009 for PCNA) (**d**). **e, f** Double immunofluorescence for Ki-67 and HNF4α in the liver 48 hours after PHx (**e**). Statistical analysis of the number of Ki-67-expressing cells (*n* = 3 independent mice, 3 fields per mouse, *p* = 7.335 × 10⁻⁵). The number of Ki-67-positive cells was normalized to the number of DAPI-stained dots (**f**). **g, h** Immunoblotting for STAT3, and P-STAT3 in the liver 24 hours after PHx (**g**). Statistical analysis of the band intensity of P-STAT3 in **g**. The band intensity of P-STAT3 was normalized to that of β-actin (*n* = 3 independent samples, *p* = 0.023) (**h**). **i** Detergent-resistant lipid rafts were isolated from adeno-LacZ or adeno-PROM1^GPI-EX1–FLAG mouse livers 48 hours after PHx. Protein expression levels of FLAG, GP130, and Flotillin-1 were determined by immunoblotting in each fraction after sucrose gradient ultracentrifugation. Scale bar = 100 μm. Two-sided student *t*-test; *p < 0.05, **p < 0.01, ***p < 0.001. Data are expressed as the mean ± SEM with individual values. Source data are provided as a Source Data file.

phosphate buffer for 30 min at 37 °C and then incubated with rat anti-PROM1 (Thermo Fisher Scientific, ebioscience, clone 13A4) overnight at 4 °C. Then, the sections were incubated with fluorescence-conjugated secondary antibody (Thermo Fisher Scientific, USA) for 1 hour at room temperature. The information of fluorescence-conjugated secondary antibodies was provided in Supplementary Table 3.

For tdTom or Ki-67 double immunofluorescence with HNF4α or CK19, heat-mediated antigen retrieval using a pressure cooker in citrate buffer (pH 6.0) was performed on frozen sections. After antigen retrieval, the sections were blocked with 2.5% normal horse serum for 30 min at room temperature. Then, the sections were incubated with rabbit (600-401-379; Rockland) or rat (TA180009; Thermo Fisher Scientific) anti-tdTom, rabbit anti-Ki-67 (12202; Cell Signaling Technology), mouse anti-HNF4α (ab41898; Abcam), and rabbit anti-CK19 (ab52625; Abcam) overnight at 4 °C and then incubated with fluorescence-conjugated secondary antibody (Thermo Fisher Scientific) for 1 hour at room temperature. The information of fluorescence-conjugated secondary antibodies was provided in Supplementary Table 3. After mounting with Fluoroshield™ with DAPI (F6057; Sigma), the images were captured using an LSM800 confocal microscope with ZEN 2009 software (Zeiss, Germany) and analyzed by ZEN 3.5 blue edition (Zeiss).

For immunofluorescence staining of cells, cells were fixed with 4% paraformaldehyde in 0.1 M sodium phosphate buffer (pH 7.4) for 15 min at room temperature. After fixation, the cells were permeabilized with 0.1% Triton X-100 for 10 min at room temperature and blocked with 5% bovine serum albumin for 30 min at room temperature. The cells were then incubated with primary antibody at their working concentrations for 1 hour at room temperature followed by incubation with fluorescence-conjugated secondary antibody (Thermo Fisher Scientific) for 1 hour at room temperature. The information of fluorescence-conjugated secondary antibodies was provided in Supplementary Table 3.

### Correlative light and electron microcopy
Correlative light and electron microscopy (CLEM) was performed as previously described[26]. The liver tissues from PHx *Prom1^f/f* and *Prom1^LKO* male mice were fixed with 4% paraformaldehyde, cryoprotected with 2.3 M sucrose (0.1 M phosphate buffer) and frozen in liquid nitrogen. The frozen tissues were cut to 1-um-thick at −100°C with Leica EM UC7 ultramicrotome. The sections were labeled at 4°C overnight using an anti-PROM1 rat monoclonal antibody (13A4, 1:200) and visualized using an Alexa Fluor 488-Fluoro Nanogold (Nanoprobes, 1:100). Cover slipped sections were detected with a confocal microscope (Zeiss, LSM700) with a differential interference contrast setting to find specific areas for later examination by electron microscopy. After coverslips had been floated off, silver enhancement was performed using HQ silver enhancement kit (Nanoprobes). After prepared for electron microscopy, areas of interest were excised and cut into 70–90 nm

thick. The samples were observed in an electron microscope (JEM 1010; JEOL, Tokyo, Japan).

### Giant plasma membrane vesicles isolation
The isolation of giant plasma membrane vesicles (GPMV) was performed according to previous reports (Levental et al.)[32,33]. The transfected cells were labeled with DiI and/or wheat germ agglutinin (5ug/ml) at 4°C for 10 min. The cells were washed twice with GPMV buffer (10 mM HEPES, 150 mM NaCl, 2 mM CaCl₂, pH7.4). The cells were incubated in GPMV buffer containing 25 mM paraformaldehyde and 2 mM dithiothreitol at 37°C for 2 hours. After incubation, GPMV containing cellular supernatant was collected into a microcentrifuge tubes. To remove remaining cell debris, the suspension was centrifuged at 100 g for 10 min. The isolated GPMVs were cooled down and observed with confocal microscopy (Zeiss, LSM700). The collected images were analyzed by ZEN 3.5 blue edition (Zeiss).

### Wheat germ agglutinin (WGA) labelling
For WGA labelling, cells were washed with phosphate-buffered saline (PBS), pH 7.4 and stained with 5ug/ml wheat germ agglutinin Alexa Flour 488 conjugate (W11261; Thermo fisher scientific) in PBS for 10 min. After labelling, the cells were washed with PBS and fixed with 4% paraformaldehyde in 0.1 M sodium phosphate buffer (pH 7.4) for 15 min at room temperature.

### Immunohistochemistry
For hematoxylin-eosin staining of liver tissues, paraffin-embedded tissues were cut to a thickness of 5 μm using a multirotary microtome (Leica). The sections were stained with hematoxylin-eosin according to a standard protocol. After mounting with synthetic mountant (Thermo Fisher Scientific), the images were captured using a light microscope (Leica).

### TUNEL assay
To analyze apoptosis in liver, terminal deoxynucleotidyl transferase dUTP nick end labeling (TUNEL) was performed according to the manufacturer's protocol (ab66110; Abcam). Briefly, frozen liver sections were fixed with 4% paraformaldehyde in PBS for 15 min at room temperature. The slides were incubated with 20ug/ml proteinase K in Tris-HCl buffer (pH 8.0, 50 mM EDTA) for 5 min at room temperature and refixed with 4% paraformaldehyde in PBS 5 min at room temperature. The slides were labeled with DNA labeling solution containing TdT enzyme and Br-dUTP for 1 hour at 37 °C. The slides were then incubated with anti-BrdU-Red antibody for 30 min at room temperature. After labelling, the slides were counter-stained with DAPI and detected by LSM700 confocal microscope (Zeiss).

### Serum IL-6 ELISA
Serum IL-6 levels were quantified using a commercial mouse IL-6 ELISA kit (RAB0308; Sigma) according to the manufacturer's instructions

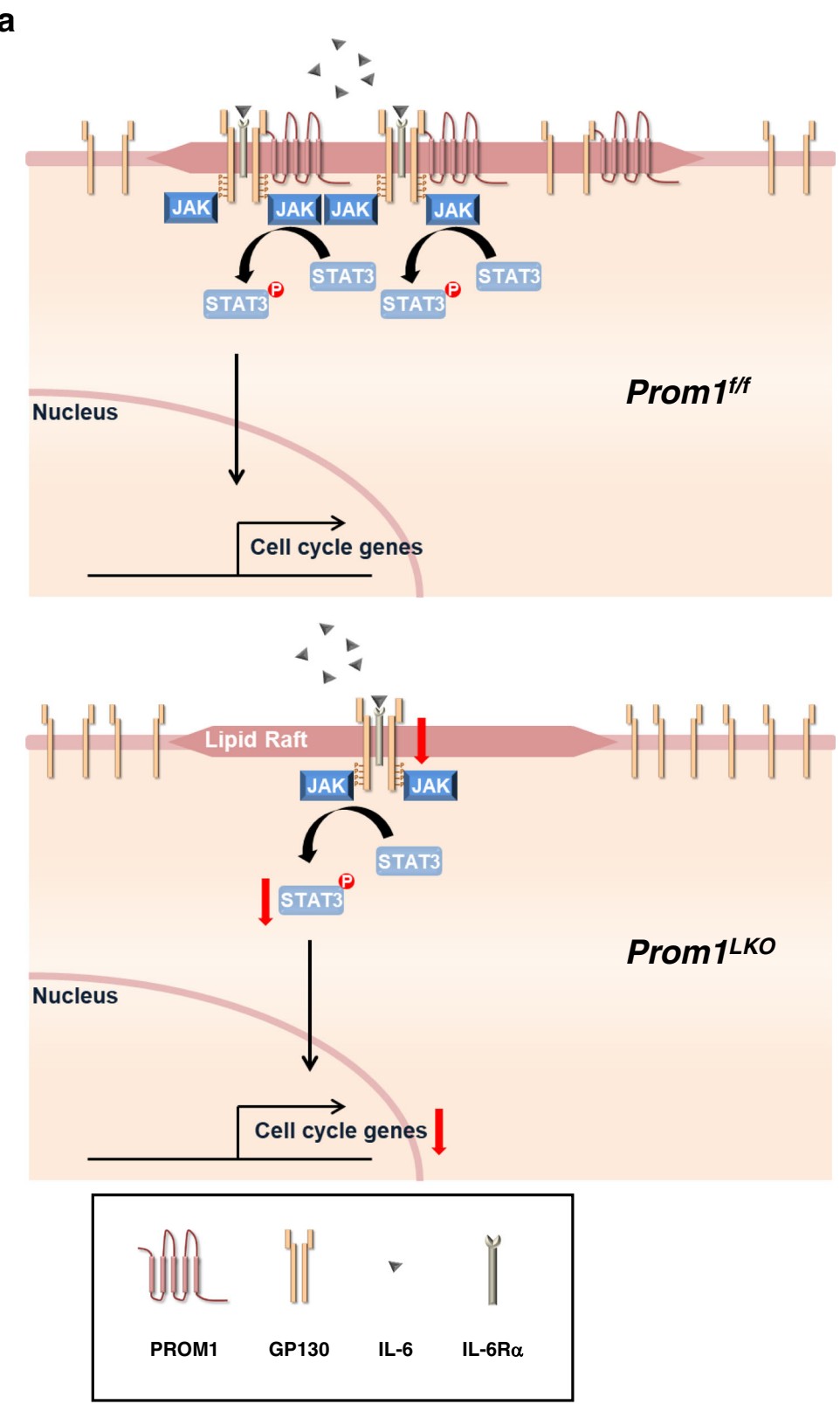

**Fig. 8 | PROM1 promotes hepatocyte proliferation through facilitating IL-6-GP130 signaling pathway in lipid rafts during liver regeneration. a** After 2/3 partial hepatectomy or CCl₄ injection, PROM1 expression increases in hepatocytes. Upregulated PROM1 recruits GP130 into lipid rafts by interacting with GP130.

PROM1/GP130 complex in lipid rafts facilitates the IL-6-GP130-STAT3 signaling pathway. Therefore, PROM1 promotes hepatocyte proliferation during liver regeneration.

and analyzed using spectra-iMAX with SoftMaz Pro V6 software (Molecular Devices, USA).

## Detergent-resistant lipid rafts isolation

The detergent-resistant lipid rafts isolation was performed as previously described[51]. To obtain detergent-resistant lipid rafts, homogenized liver tissues were lysed with buffer containing 1% Brij-35, 25 mM HEPES pH 6.5, 150 mM NaCl, 1 mM EDTA, and protease inhibitor cocktail (P3100; Gendepot) on ice for 30 min. Then, the lysates were subjected to discontinuous sucrose gradient (40, 35, and 5%) ultracentrifugation using SW55Ti rotor or SW41Ti rotor (28,7000 × g) for 18 hours at 4 °C. After ultracentrifugation, the sucrose solutions were fractionated into 10-12 fractions. A cloudy band corresponding to the lipid rafts was collected at the interface between the 35 and 5% sucrose solutions and confirmed by immunoblotting for Flotillin-1 as a lipid raft marker.

## Plasmid construction and transient transfection

Deletion mutants of FLAG-tagged human PROM1 transcript variant 2 (PROM1-FLAG) were generated by reverse PCR as previously described[26]. FLAG-tagged GPI-anchored PROM1-EX1 was generated by the DNA assembly method (#E2621, NEB, Ipswich, MA, USA). The GPI-anchor signal sequence from pCAG:GPI-GFP (#32601, Addgene) was added at the C-terminus of PROM1-EX1 (1-99)-FLAG. His-tagged GP130 was generated by adding a 6×His tag sequence at the C-terminus of the GP130 CDS obtained from the cDNA library of HEK293 cells.

DNA transfection was performed using Lipofectamine 3000 reagent (Invitrogen, USA) according to the manufacturer's instructions.

## Statistics and reproducibility

The number of mice used in each experiment was determined based on preliminary experiments in the same model. Immunofluorescence images and immunoblotting band intensities were quantified using ImageJ 1.52i (NIH) or Photoshop CD5 (Adobe) software. The images used for statistics contained more than ~250 cells per field and were taken from a minimum of 3-5 fields per sample. Data are expressed as the mean ± SEM with individual values using Graphpad Prism 6 software. Sample numbers are indicated in the figure legends. A two-tailed Student's $t$-test was used to calculate the $p$ values. Significance levels were *$p < 0.05$; **$p < 0.01$; ***$p < 0.001$; and n.s, nonsignificant. A $p$ value $< 0.05$ was considered statistically significant. Each experiment was repeated independently three times with similar results.

## Reporting summary

Further information on research design is available in the Nature Research Reporting Summary linked to this article.

# Data availability

Source data containing uncropped blots and raw data for all plots are provided with this paper.

All other data supporting this study are available within the paper and its supplementary information. Source data are provided with this paper.

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

## Acknowledgements

We thank all members of our laboratory for their supports and intellec-tual inputs during the preparation of this manuscript.

## Author contributions

M.-S.B., D.-M.Y., M.L., S.-J.J., J.-W.L., H.-C.K., H.L., H.L.K., A.K., and J.S.K. performed the experiments; J.-H.H., S.-H.K., J.-S.L. and Y.-G.K. designed the experiments and analyzed the data; and M.-S.B., D.-M.Y. and Y.-G.K. wrote the manuscript.

## Funding

This work was supported by grants from the National Research Foun-dation of Korea awarded to; Y.-G. Ko (R1509597 and R20000552) and J.-S. Lee (R1A5A2031612).

## Competing interests

The author declares no competing interests.
