## [Peer Review File · Nature Communications]

Central role of Prominin-1 in lipid rafts during liver regenerationREVIEWER COMMENTS

Reviewer #1 (Remarks to the Author):

Prominin-1 (Prom1) was classically defined as a raft associated protein in early studies on the basis of its association with detergent resistant domains. It is expressed in cancer stem cells as well as other stem cell populations. In this work, Bahn, Yu et al examine Prom1's role in liver regeneration. Using several models to induce liver damage and the regeneration process, they show that Prom1 levels increase during regeneration and liver-specific Prom1 knockout mice exhibit slowed regeneration compared to wild type mice, and provide evidence Prom1 regulates IL-6 signaling. They go on to report that Prom1 appears to increase IL-6 signaling by recruiting the interleukin-6 signal transducer glycoprotein 130 (GP130) to detergent resistant membrane fractions, show that the two proteins can co-IP, identify the first extracellular domain of Prom1 as the region required to co-IP with GP130, and demonstrate that in the absence of Prom1 expression in the liver that GP130 remains in detergent soluble membranes. Finally, they design a chimeric protein consisting of the first extracellular domain of Prom1 attached to a GPI-anchor, and find it is both capable of interacting with GP130 and activating signaling when overexpressed in the Prom1 liver knockout mice. The authors conclude that recruitment of GP130 into rafts by Prom1 during regeneration is required to activate STAT3 signaling by IL-6.

In general, this study appears to have been well performed and results are consistent with a key role for Prom1 in the pathways described here. I am not an expert on either liver regeneration or Prominin-1 but presume that those that work on these topics would find this study to be of interest. My comments and questions focus primarily on the lipid raft angle of the story.

1. The authors use detergent resistant membranes (DRMs) as a tool to analyze whether Prom1 and GP130 localize to rafts. Given that the bulk of the experiments done here involve intact tissue, this makes sense. That being said, the field now recognizes that this approach is limited and that DRMs cannot be equated with rafts (1-3). Knowledge about mechanisms that target specific proteins to rafts have also evolved also recent years, including the development of models that predict how the structure of single-pass transmembrane proteins regulates their affinity for ordered domains (4). These insights have emerged from a more up-to-date raft model, giant plasma membrane vesicles or GPMVs (5,6). Notably, only a few examples of multi-pass transmembrane proteins that associate with the raft domain of GPMVs have been reported so far (7,8).

Given that the authors are focusing on the connections between Prom1 and rafts as being of critical importance in the pathway they are studying, they should ideally carry out experiments to test their proposed model using GPMVs. Happily, the authors have already demonstrated that they are able to detect (presumed) interactions between Prom1 and GP130 in HEK cells using co-IP experiments. It would thus be very informative to test whether Prom1 indeed associates with the raft phase of GPMVs derived from HEK cells (or another cell line). If the proposed model is indeed correct, one would expect to find that Prom1 itself (and GPI-Prom1-EX1) would show a propensity to localize in the raft phase of GPMVs; GP130 should associate primarily with non-raft domains in GPMVs in the absence of Prom1; and GP130 should be recruited to the raft phase when co-expressed with Prom1 or GPI-Prom1-EX1.

I recognize these experiments may be challenging for the authors to do, but think that such studies would increase the general interest of this work by providing more definitive evidence that Prom1 is indeed a bona fide raft-associated multi-pass transmembrane protein and is capable of recruiting GP130 to rafts using these more timely and rigorous criteria. If they are unable to do them, they should at minimum discuss potential raft targeting mechanisms of Prom1 and GP130 in light of the current state of knowledge outlined above.

2. The model proposed in Figure 8 implies that GP130 is located outside of rafts (DRMs) in the intact liver, prior to upregulation of Prom1 levels in response to liver damage. Unless I missed it, I did not see any experiments where this was examined in control livers. Have the authors tested

whether this is the case?

3. At least one previous study has shown that GP130 can be actively recruited from non-raft domains into DRMs (9). Others have reported GP130 can be found in DRMs (10,11). These and related works should be cited and discussed.

Other comments.

a. For a paper appearing in Nature Communications, the reader expects the introduction and discussion to clearly state why the results of the study are of broad interest, including the major outstanding questions that the study answers. These points did not come across clearly (at least to this reader). I suggest reworking the text along these lines.

b. Figure 1b shows multiple lanes under the heading of PHx that are blotted for Prom1. It is unclear what the different lanes correspond to. Are these from different mice? This should be clarified.

c. In many figures, the bands shown in the Western blots appear to be saturated. Can the authors please clarify whether this is the case?

d. In several places in the manuscript the text states that experiments were designed to "prove" hypotheses. To my understanding, one cannot prove a hypothesis; hypotheses can only be disproven. Similarly, results cannot prove a hypothesis; they can however be consistent with the predictions of a hypothesis. These statements in the text should be rephrased accordingly.

e. The authors performed immunoprecipitation experiments to test for possible interactions between GP130 and Prom1. While the results of these experiments are consistent with the possibility that they interact, it is not possible to exclude the possibility from this type of experiment that they are part of a larger complex and do not directly interact with one another. This needs to be clarified in the text.

f. Several of the co-IP experiments were carried out in HEK cells expressing epitope tagged Prom1 and GP130. It would be useful to know if these cells express any endogenous Prom1 or GP130.

g. Were any experiments done to confirm that the Prom1-GPI-EXi construct is GPI anchored as expected?

h. The gender of the mice used in the study should be indicated.

i. A citation should be provided for the protocol used to isolate detergent-resistant lipid rafts.
References

1. Lingwood, D., and Simons, K. (2007) Detergent resistance as a tool in membrane research. *Nat Protoc* 2, 2159-2165
2. Simons, K., and Gerl, M. J. (2010) Revitalizing membrane rafts: new tools and insights. *Nat Rev Mol Cell Biol* 11, 688-699
3. Levental, I., Levental, K. R., and Heberle, F. A. (2020) Lipid rafts: Controversies resolved, mysteries remain. *Trends Cell Biol* 30, 341-353
4. Lorent, J. H., Diaz-Rohrer, B., Lin, X., Spring, K., Gorfe, A. A., Levental, K. R., and Levental, I. (2017) Structural determinants and functional consequences of protein affinity for membrane rafts. *Nat Commun* 8, 1219
5. Levental, K. R., and Levental, I. (2015) Isolation of giant plasma membrane vesicles for evaluation of plasma membrane structure and protein partitioning. *Methods Mol Biol* 1232, 65-77
6. Sezgin, E., Kaiser, H. J., Baumgart, T., Schwille, P., Simons, K., and Levental, I. (2012) Elucidating membrane structure and protein behavior using giant plasma membrane vesicles. *Nat Protoc* 7, 1042-1051
7. Castello-Serrano, I., Lorent, J. H., Ippolito, R., Levental, K. R., and Levental, I. (2020) Myelin-Associated MAL and PLP Are Unusual among Multipass Transmembrane Proteins in Preferring Ordered Membrane Domains. *J Phys Chem B* 124, 5930-5939
8. Marinko, J. T., Kenworthy, A. K., and Sanders, C. R. (2020) Peripheral myelin protein 22 preferentially partitions into ordered phase membrane domains. *Proc Natl Acad Sci U S A* 117, 14168-14177
9. Port, M. D., Gibson, R. M., and Nathanson, N. M. (2007) Differential stimulation-induced receptor localization in lipid rafts for interleukin-6 family cytokines signaling through the gp130/leukemia inhibitory factor receptor complex. *J Neurochem* 101, 782-793
10. Yanagisawa, M., Nakamura, K., and Taga, T. (2004) Roles of lipid rafts in integrin-dependent

adhesion and gp130 signalling pathway in mouse embryonic neural precursor cells. *Genes Cells* 9, 801-809

11. Buk, D. M., Waibel, M., Braig, C., Martens, A. S., Heinrich, P. C., and Graeve, L. (2004) Polarity and lipid raft association of the components of the ciliary neurotrophic factor receptor complex in Madin-Darby canine kidney cells. *J Cell Sci* 117, 2063-2075

Reviewer #2 (Remarks to the Author):

Hepatology Review:

Authors studied the central role of Prominin-1 in lipid rafts during liver regeneration. The main point of this manuscript I understand was that "PROM1 is upregulated in hepatocytes during liver regeneration and upregulated PROM1 recruits GP130 to lipid rafts and activates the IL6-GP130-STAT3 axis".

Authors investigated the role of PROM1 in lipid rafts during liver regeneration. The expression of PROM1 increased during liver regeneration after PHx or CCl4 injection. Hepatocyte proliferation and liver regeneration were attenuated in liver-specific Prom1 knockout (Prom1LKO) mice compared to wild-type (Prom1f/f) mice. PROM1 interacted with the IL6 GP130 and confined GP130 to lipid rafts. Therefore, STAT3 by IL-6 was activated. The overexpression of GPI-anchored first extracellular domain of PROM1 (PROM1GPIEX1) rescued the proliferation of hepatocytes and liver regeneration in Prom1LKO mice. PROM1 is upregulated in hepatocytes during liver regeneration, and upregulated PROM1 recruits GP130 into lipid rafts and activates the IL6-GP130-STAT3 pathways. Authors concluded that PROM1 regulates lipid rafts during liver regeneration and authors propose therapeutic applications of liver transplantation. Language was well written with many good figures as well. Molecular mechanisms, however, are not clearly defined.

Main points:

1. What about the H&E staining results for Figure 7E? Does the H&E staining result show that hepatocyte proliferation is increased when PROM1GPI-EX1 is overexpressed compared to LacZ overexpression?
2. Have you measured the level of IL-6 in PROM1-LKO mice after PHx / CCl4 treatment? If the activation of the IL6-GP130-STAT3 axis is decreased due to PROM1-LKO, does it affect the expression of various growth factors or cytokines that are increased during liver regeneration?
3. Sham control was not described in the methods or text. An explanation on which type of mice this was derived from would be useful.
4. N=3 mice (out of N=9) for PHx representative mice (Fig. 1B) show no or little CD133 expression. Why are 33% of these mice do not show this expression since the other 6 mice do?

These are the criticism that I have for this manuscript.

Minor Comments:

1. In Fig 2F, expression of DAPI and Ki67 should be enhanced as the frequency of Ki67 looks greater from sham model compared to PDX mice. IRS or some other form of quantification should be shown for at least n=3 mice for each IHC image.
2. mRNA expression levels from Figure 1A shows expression after 24 and 120 hours after PHx. It would be good to see Western blot representative blots of at least n=3 for each of these groups as only 48 hours after PHx was shown (Fig. 1B).

Reviewer #3 (Remarks to the Author):

The authors of the manuscript entitled "Central role of Prominin-1 in lipid rafts during liver regeneration" describe the implication of the stem cell marker prominin-1 and lipid rafts during the process of liver regeneration, a phenomenon that relies on the interaction of the N-terminal domain of prominin-1 with the interleukin-6 signal transducer glycoprotein 130. Although this manuscript presents interesting data, many major concerns need to be clarified, including the role of lipid rafts in these processes.

Major Points

1. Please provide a high power view of the immunofluorescence images to appreciate the subcellular localization of prominin-1 in the cells of interest and describe in detail its subcellular localization. For example, where is it located in the hepatocytes of the PHx liver? Immunogold electron microscopy could help define its localization. As mentioned in the introduction, prominin-1 is found in membrane protrusions. Please provide data on prominin-1 localization in primary hepatocytes under the different conditions used.
2. Do prominin-1 knockout mice show a phenotype related to a liver defect? This information must be provided.
3. In the Results or Methods sections, the authors should explain the use of Brij-35 as a detergent employed for the isolation of lipid rafts. Classically, Triton X-100 is used. Indeed, GP130 is insoluble in Triton X-100 (Ref. 19), whereas prominin-1 is soluble in Triton X-100, suggesting that they do not localize in the same type of lipid rafts (Ref. 9). The authors should comment on this, and perhaps address these questions by comparing with data using Triton X-100 instead of Brij-35. Was co-localization of prominin-1 and GP130 observed under the different conditions used, and with the different prominin-1 deletion mutants?
4. Is the interaction of prominin-1 and GP130 dependent on lipid rafts? According to the methods, the materials prepared for immunoisolation experiments are based on detergent lysates obtained after cell solubilization with the detergent NP-40, which will completely solubilize prominin-1, suggesting that lipid rafts are not involved in this cross-talk. This should be clarified. Does membrane cholesterol play a role in these interactions?
5. Do prominin-1 and GPI-anchored PROM1-EX1 co-localize in the same type of lipid rafts? According to Ref. 9, they will not. This question needs to be addressed biochemically and with respect to their co-localization in the cells of interest.

Other Points

1. In the Introduction section, the authors should describe the normal expression pattern of prominin-1 in liver (human and mouse) including canals of Hering and bile ducts (e.g., Immervoll et al. BMC Cancer 2008 PMID: 18261235; Karbanová et al. J Histochem Cytochem 2008 PMID: 18645205). These papers should be cited instead of the review of Glumac and Lebeau (Ref. 11) which does not provide such detailed information. It is essential to give credit to genuine contributors. Citation of manuscripts that have not yet been peer-reviewed should not be encouraged, as is the case in bioRxiv (e.g., Ref. 16).
2. In Supplementary Table 2, authors should provide the clone name and/or catalog number of the antibody. A reference for each must be provided as validation.
3. In the Methods section (Immunofluorescence staining), is the rat anti-PROM1 (Thermo Fisher Scientific) the same as described in Supplementary Table 2 and acquired from eBioscience? Are we talking about the rat monoclonal antibody 13A4 used to identify prominin-1 (Weigmann et al. PNAS 1997 PMID: 9356465)? These details are important given that anti-prominin-1 antibodies have been the subject of much controversy.

4. In the histograms, please provide individual values and not just the mean and S.E.M.
5. The uncropped blots should be provided in the Supplementary Materials. Does any other prominin-1-related immunoreactive bands appear on the whole blot (e.g., Figure 2C)? Molecular mass marker should be indicated.
6. What are the different lanes of the blot (PHx) shown in Figure 1B? Different independent samples? The variability seems to be very high. How do the authors explain this?
7. Does apoptosis occur under the different conditions?
8. Figure 4A, C, as an internal control, prominin-1 blots should be shown.
9. It is indicated in the Methods that variant 2 of the human PROM1 transcript is used. Please provide the source of this nomenclature. According to Fargeas et al. (Tissue Antigens 2007, PMID:17498271), the human s2 variant will be 865 amino acids in length, not 856 residues as shown in Figure 6A. The length of 856 amino acids will correspond to the s1 variant.

Reviewer #1 (Remarks to the Author):

Prominin-1 (Prom1) was classically defined as a raft associated protein in early studies on the basis of its association with detergent resistant domains. It is expressed in cancer stem cells as well as other stem cell populations. In this work, Bahn, Yu et al examine Prom1's role in liver regeneration. Using several models to induce liver damage and the regeneration process, they show that Prom1 levels increase during regeneration and liver-specific Prom1 knockout mice exhibit slowed regeneration compared to wild type mice, and provide evidence Prom1 regulates IL-6 signaling. They go on to report that Prom1 appears to increase IL-6 signaling by recruiting the interleukin-6 signal transducer glycoprotein 130 (GP130) to detergent resistant membrane fractions, show that the two proteins can co-IP, identify the first extracellular domain of Prom1 as the region required to co-IP with GP130, and demonstrate that in the absence of Prom1 expression in the liver that GP130 remains in detergent soluble membranes. Finally, they design a chimeric protein consisting of the first extracellular domain of Prom1 attached to a GPI-anchor, and find is both capable of interacting with GP130 and activating signaling when overexpressed in the Prom1 liver knockout mice. The authors conclude that recruitment of GP130 into rafts by Prom1 during regeneration is required to activate STAT3 signaling by IL-6.

In general, this study appears to have been well performed and results are consistent with a key role for Prom1 in the pathways described here. I am not an expert on either liver regeneration or Prominin-1 but presume that those that work on these topics would find this study to be of interest. My comments and questions focus primarily on the lipid raft angle of the story.

Comment 1: The authors use detergent resistant membranes (DRMs) as a tool to analyze whether Prom1 and GP130 localize to rafts. Given that the bulk of the experiments done here involve intact tissue, this makes sense. That being said, the field now recognizes that this approach is limited and that DRMs cannot be equated with rafts (1-3). Knowledge about mechanisms that target specific proteins to rafts have also evolved also recent years, including the development of models that predict how the structure of single-pass transmembrane proteins regulates their affinity for ordered domains (4). These insights have emerged from a more up-to-date raft model, giant plasma membrane vesicles or GPMVs (5,6). Notably, only a

few examples of multi-pass transmembrane proteins that associate with the raft domain of GPMVs have been reported so far (7,8).

Given that the authors are focusing on the connections between Prom1 and rafts as being of critical importance in the pathway they are studying, they should ideally carry out experiments to test their proposed model using GPMVs. Happily, the authors have already demonstrated that they are able to detect (presumed) interactions between Prom1 and GP130 in HEK cells using co-IP experiments. It would thus be very informative to test whether Prom1 indeed associates with the raft phase of GPMVs derived from HEK cells (or another cell line). If the proposed model is indeed correct, one would expect to find that Prom1 itself (and GPI-Prom1-EX1) would show a propensity to localize in the raft phase of GPMVs; GP130 should associate primarily with non-raft domains in GPMVs in the absence of Prom1; and GP130 should be recruited to the raft phase when co-expressed with Prom1 or GPI-Prom1-EX1.

I recognize these experiments may be challenging for the authors to do, but think that such studies would increase the general interest of this work by providing more definitive evidence that Prom1 is indeed a bona fide raft-associated multi-pass transmembrane protein and is capable of recruiting GP130 to rafts using these more timely and rigorous criteria. If they are unable to do them, they should at minimum discuss potential raft targeting mechanisms of Prom1 and GP130 in light of the current state of knowledge outlined above.

Answer 1: In order to examine whether PROM1 is recruited into the raft phase of GPMV, HEK 293 cells were overexpressed with PROM1-GFP or GPI-GFP and stained with DiI for a non-raft marker. GPMVs were isolated from the cells using PFA and DTT according to previous reports (Levental et al)^{1, 2}, cooled down and observed by confocal microscopy. As shown in below Fig. R1, GPMVs were successfully isolated and partitioned to two phases because GPI-GFP and DiI signals were not co-localized with each other. Unexpectedly, PROM1-GFP was co-localized with DiI in same phase. Because wheat germ agglutinin (WGA) is known to bind glycoproteins³ and co-localize with PROM1 microdomain⁴, we tried to investigate whether WGA-labeled membranes were located at the raft phase in GPMV. WGA existed in non-raft phase with PROM1-tdTom in below Fig. R1. Our results suggest that PROM1 and WGA-labeled proteins were located to non-raft phase in GPMV. Although WGA-binding glycoproteins including PROM1 are detergent-resistant membrane protein and have a typical punctate staining pattern for lipid raft proteins⁴, they exist at non-raft phase in GPMV.

Lipid rafts are defined as a membrane domain resistant to non-ionic detergent because they are tightly packed with glycosphingolipids and cholesterol^{5, 6, 7}. The tight packaging of glycosphingolipids and cholesterol also induces phase partitioning in GPMV^{2, 8}. However, phase partitioning in GPMV is different from DRM for many multi-span transmembrane proteins due to the disruption of cytoskeleton network, lipid bilayer asymmetry, and protein-protein interaction in GPMV⁹. In previous our report¹⁰, PROM1 interacts with cortical actin by radixin. Since cortical actin might be disrupted during GPMV isolation, it seems that we failed to observe raft recruitment of PROM1 in GPMV.

Because PROM1 was not found in raft phase of GPMV, we could not proceed the observation of the phase separation of GP130 in GPMV. Alternatively, we determined cellular GP130 localization in the presence of PROM1. GP130-His was overexpressed in HEK293 cells along with or without PROM1-FLAG. The localization of GP130 and PROM1 was determined by immunofluorescence after WGA labeling. As shown in Fig. 5G, in the absence of PROM1, GP130 was mainly found in intracellular compartments, which was not labeled with WGA. In the presence of PROM1, GP130 was co-localized with PROM1 in the plasma membrane, which was labeled with WGA. These data suggest that PROM1 recruits GP130 to the WGA-labeled membrane in HEK 293 cells.

Figure R1. PROM1 is localized in the non-raft phase of GPMV. GPMVs were isolated from HEK 293 cells expressing GPI-GFP, PROM1-GFP or -tdTom. DiI (as a non-raft marker) and WGA (as a glycoprotein marker) staining was performed before GPMV isolation. Scale bar = 5μm.

Comment 2: The model proposed in Figure 8 implies that GP130 is located outside of rafts (DRMs) in the intact liver, prior to upregulation of Prom1 levels in response to liver damage. Unless I missed it, I did not see any experiments where this was examined in control livers. Have the authors tested whether this is the case?

Answer 2: To clarify GP130 raft localization in sham livers, we isolated lipid rafts using Brij-35 from 8-week-old male *Prom1^{ff}* and *Prom1^{LKO}* mice livers after sham operation. As shown in Fig. 5A, PROM1 was expressed in lipid rafts in *Prom1^{ff}* liver. However, GP130 was expressed in non-rafts in both *Prom1^{ff}* and *Prom1^{LKO}* livers. Because PROM-1 positive

hepatocytes accounted for ~1% of total hepatocytes in sham liver (Fig. 1F and G), GP130 was present in non-rafts.

Comment 3: At least one previous study has shown that GP130 can be actively recruited from non-raft domains into DRMs (9). Others have reported GP130 can be found in DRMs (10,11). These and related works should be cited and discussed.

Answer 3: GP130 is recruited from non-raft to lipid rafts after ciliary neurotrophic factor (CNTF) treatment in neural cells (IMR-32 cells)¹¹ whereas it is always found in lipid rafts independent on ligand activation in mouse embryonic neural precursor cells¹², Madin-Darby canine kidney cells¹³, and Hep3B cells¹⁴. Our data showed that GP130 was found in lipid rafts in PROM1-overexpressing *Prom1*^{LKO} sham liver (Fig.5C), indicating GP130 localization in lipid rafts is dependent on PROM1 but not IL-6 activation. We added this context in the “Discussion” section.

Other comments.

Comment a: For a paper appearing in Nature Communications, the reader expects the introduction and discussion to clearly state why the results of the study are of broad interest, including the major outstanding questions that the study answers. These points did not come across clearly (at least to this reader). I suggest reworking the text along these lines.

Answer a: For a general introduction to liver regeneration, we added paragraphs with underline to the “Introduction” and “Discussion” sections as follows (References were cited in the main text):

In the Introduction section

“The liver is a pivotal organ for maintaining homeostasis by regulating metabolism, drug detoxification and bile transportation. Hepatocytes, the major parenchymal cells in the liver, could be damaged by various factors such as surgical operation, alcohol, virus and chemicals, which leads to a decrease in liver mass. To maintain homeostasis, the liver has a unique

capability to recover its original mass. Many studies for cytokines or growth factors have tried to contribute therapeutic approaches to promote liver regeneration. For example, there are antibody, agonist or antagonist therapy targeting for specific signaling pathway in liver regeneration. In addition, some studies have attempted to use a cell population with stemness for a cell therapy. Understanding the molecular mechanisms in liver regeneration is important for application in the field of liver disease therapy.”

In the Discussion section

“PROM1 is well known as a marker for cancer stem cells and normal stem cells. Recent studies have revealed its ability to regulate various cellular signal transduction pathways by interacting with PI3K, HDAC6, radixin, and SMAD7. Indeed, PROM1-deficiency leads to the prevention of glucagon-induced gluconeogenesis via inactivating the function of radixin as A kinase-anchoring protein (AKAP), and aggravation of bile duct ligation (BDL)-induced liver fibrosis via SMAD7 degradation, indicating that PROM1 has different functions in the liver. Here, we demonstrated that PROM1 is also necessary for regulating IL-6 signaling during liver regeneration. We found that the expression of PROM1 dramatically increased in hepatocytes during liver regeneration after PHx or CCl₄ injection. Hepatocellular PROM1 facilitated the IL-6 signaling pathway by interacting with GP130 in lipid rafts. As a result, PROM1 promoted the proliferation of hepatocytes during liver regeneration (Fig. 8). Thus, this study is the first to elucidate the function of PROM1 in liver regeneration and is expected to provide a deeper understanding of liver regeneration and liver transplantation therapy.”

Comment b: Figure 1B shows multiple lanes under the heading of PHx that are blotted for

Prom1. It is unclear what the different lanes correspond to. Are these from different mice? This should be clarified.

Answer b: Fig. 1B shows immunoblots for PROM1 in wild type sham liver (N=2) and 48h PHx liver (N=9). Left two lanes represent sham liver samples and the next nine lanes represent 48h PHx liver specimen of 8-week-old wild-type male mice. Each different lane corresponds to a different mouse liver specimen. To avoid confusion, we have marked the number of mice above the lanes in Fig.1B.

Comment c: In many figures, the bands shown in the Western blots appear to be saturated. Can the authors please clarify whether this is the case?

Answer c: At the reviewer's request, the western blot data with saturated bands were replaced.

Comment d: In several places in the manuscript the text states that experiments were designed to "prove" hypotheses. To my understanding, one cannot prove a hypothesis; hypotheses can only be disproven. Similarly, results cannot prove a hypothesis; they can however be consistent with the predictions of a hypotheses. These statements in the text should be rephrased accordingly.

Answer d: We appreciated the reviewer's comment. The statements "to prove this hypothesis" in the manuscript have been rephrased to "to examine the possibility".

Comment e: The authors performed immunoprecipitation experiments to test for possible interactions between GP130 and Prom1. While the results of these experiments are consistent with the possibility that they interact, it is not possible to exclude the possibility from this type of experiment that they are part of a larger complex and do not directly interact with one another. This needs to be clarified in the text.

Answer e: At the reviewer's request, we rephrased for the interaction between GP130 and PROM1 as the followings in the "Result" section: "All these data indicate that PROM1 and GP130 forms a complex, which is important for the raft localization of GP130."

Comment f: Several of the co-IP experiments were carried out in HEK cells expressing epitope

tagged Prom1 and GP130. It would be useful to know if these cells express any endogenous PROM1 or GP130.

Answer f: To confirm endogenous PROM1 and GP130 level, we tried to detect PROM1 and GP130 in HEK 293 cells using anti-PROM1 and anti-GP130 antibodies. HepG2 cell lines, which express PROM1 and GP130, were used as a positive control. As shown in below Figure R2, HEK 293 cells don't express endogenous PROM1 and express low levels of endogenous GP130.

Figure R2. The expression of endogenous PROM1 and GP130 in HEK 293 cells. PROM1-FLAG and GP130-His were overexpressed in HEK 293 cells. PROM1, GP130 and β -actin were determined by immunoblottings.

Comment g: Were any experiments done to confirm that the Prom1-GPI-EXi construct is GPI anchored as expected?

Answer g: Because phosphatidylinositol-specific phospholipase C (PI-PLC) is well known to release GPI-anchored proteins from plasma membrane, we observed the release of GPI-anchored PROM1-EX1 from HEK 293 cells overexpressing FLAG-tagged PROM1^{GPI-EX1} after PI-PLC treatment. Cell surface immunostaining with a Flag antibody revealed that the membrane expression of GPI-anchored PROM1-EX1 was decreased after PI-PLC treatment (below Figure R3A). In addition, immunoblotting showed that PROM1^{GPI-EX1} protein level was decreased in whole cell lysates, whereas was increased in culture medium after PI-PLC treatment (below Figure R3B). pCAG: GPI-GFP (addgene #32601) construct was used as a positive control.

Figure R3. PROM1^{GPI-EX1} possesses a GPI-anchor. HEK 293 cells were transfected with GPI-GFP or FLAG-tagged PROM1^{GPI-EX1} vector. After 24 hours, PI-PLC (1U/ml) was treated for 1 hour at 37°C. GPI-GFP was used as a positive control. (A) The surface immunofluorescence for GFP or FLAG after PI-PLC treatment. (B) GFP, FLAG, and β-actin in whole cell lysates (WCL) and media after PI-PLC treatment were determined by immunoblottings. Scale bar = 10um.

Comment h: The gender of the mice used in the study should be indicated.

Answer h: We have specified in the "Materials and methods" section and in each figure legend that the gender of all mice used in each experiment was male.

Comment i: A citation should be provided for the protocol used to isolate detergent-resistant lipid rafts.

Answer i: The citation has been provided for the lipid raft isolation protocol in the “Materials and methods” section¹⁵.

References (Reviewer #1’s references)

1. Lingwood, D., and Simons, K. (2007) Detergent resistance as a tool in membrane research. *Nat Protoc* 2, 2159-2165
2. Simons, K., and Gerl, M. J. (2010) Revitalizing membrane rafts: new tools and insights. *Nat Rev Mol Cell Biol* 11, 688-699
3. Levental, I., Levental, K. R., and Heberle, F. A. (2020) Lipid rafts: Controversies resolved, mysteries remain. *Trends Cell Biol* 30, 341-353
4. Lorent, J. H., Diaz-Rohrer, B., Lin, X., Spring, K., Gorfe, A. A., Levental, K. R., and Levental, I. (2017) Structural determinants and functional consequences of protein affinity for membrane rafts. *Nat Commun* 8, 1219
5. Levental, K. R., and Levental, I. (2015) Isolation of giant plasma membrane vesicles for evaluation of plasma membrane structure and protein partitioning. *Methods Mol Biol* 1232, 65-77
6. Sezgin, E., Kaiser, H. J., Baumgart, T., Schwille, P., Simons, K., and Levental, I. (2012) Elucidating membrane structure and protein behavior using giant plasma membrane vesicles. *Nat Protoc* 7, 1042-1051
7. Castello-Serrano, I., Lorent, J. H., Ippolito, R., Levental, K. R., and Levental, I. (2020) Myelin-Associated MAL and PLP Are Unusual among Multipass Transmembrane Proteins in Preferring Ordered Membrane Domains. *J Phys Chem B* 124, 5930-5939
8. Marinko, J. T., Kenworthy, A. K., and Sanders, C. R. (2020) Peripheral myelin protein 22 preferentially partitions into ordered phase membrane domains. *Proc Natl Acad Sci U S A* 117, 14168-14177

9. Port, M. D., Gibson, R. M., and Nathanson, N. M. (2007) Differential stimulation-induced receptor localization in lipid rafts for interleukin-6 family cytokines signaling through the gp130/leukemia inhibitory factor receptor complex. *J Neurochem* 101, 782-793
10. Yanagisawa, M., Nakamura, K., and Taga, T. (2004) Roles of lipid rafts in integrin-dependent adhesion and gp130 signalling pathway in mouse embryonic neural precursor cells. *Genes Cells* 9, 801-809
11. Buk, D. M., Waibel, M., Braig, C., Martens, A. S., Heinrich, P. C., and Graeve, L. (2004) Polarity and lipid raft association of the components of the ciliary neurotrophic factor receptor complex in Madin-Darby canine kidney cells. *J Cell Sci* 117, 2063-2075

Reviewer #2 (Remarks to the Author): Hepatology Review:

Hepatology Review:

Authors studied the central role of Prominin-1 in lipid rafts during liver regeneration. The main point of this manuscript I understand was that "PROM1 is upregulated in hepatocytes during liver regeneration and upregulated PROM1 recruits GP130 to lipid rafts and activates the IL6-GP130-STAT3 axis".

Authors investigated the role of PROM1 in lipid rafts during liver regeneration. The expression of PROM1 increased during liver regeneration after PHx or CCl4 injection. Hepatocyte proliferation and liver regeneration were attenuated in liver-specific Prom1 knockout (Prom1LKO) mice compared to wild-type (Prom1f/f) mice. PROM1 interacted with the IL6 GP130 and confined GP130 to lipid rafts. Therefore, STAT3 by IL-6 was activated. The overexpression of GPI-anchored first extracellular domain of PROM1 (PROM1GPIEX1) rescued the proliferation of hepatocytes and liver regeneration in Prom1LKO mice. PROM1 is upregulated in hepatocytes during liver regeneration, and upregulated PROM1 recruits GP130 into lipid rafts and activates the IL6-GP130-STAT3 pathways. Authors concluded that PROM1 regulates lipid rafts during liver regeneration and authors propose therapeutic applications of liver transplantation. Language was well written with many good figures as well. Molecular mechanisms, however, are not clearly defined.

Main points:

Comment 1. What about the H&E staining results for Figure 7E? Does the H&E staining result show that hepatocyte proliferation is increased when PROM1GPI-EX1 is overexpressed compared to LacZ overexpression?

Answer 1: To observe mitotic cells in LacZ and PROM1^{GPI-EX1}-overexpressed *Prom1*^{LKO} livers, H&E staining was performed. As shown in Fig. S4, the number of mitotic hepatocytes increased in Ad-PROM1^{GPI-EX1} livers compared in Ad-LacZ livers after PHx.

Comment 2. Have you measured the level of IL-6 in PROM1-LKO mice after PHx / CCl4 treatment? If the activation of the IL6-GP130-STAT3 axis is decreased due to PROM1-LKO, does it affect the expression of various growth factors or cytokines that are increased during

liver regeneration?

Answer 2: We showed the level of serum IL-6 in PHx model (Fig.4E). We also performed serum IL-6 ELISA in CCl₄ model at the reviewer's request. As shown in Fig.4E, PROM1 deficiency did not change the level of serum IL-6 after CCl₄ injection. In addition, to confirm whether PROM1 deficiency affect the level of other growth factors, we measured *Egf* and *Hgf* mRNA expression by qRT-PCR. PROM1 deficiency did not change the expression of *Egf* and *Hgf* after both PHx and CCl₄ injection (Fig.S2).

Comment 3. Sham control was not described in the methods or text. An explanation on which type of mice this was derived from would be useful.

Answer 3: The explanation for sham operation has been added in the "Materials and methods" section.

Comment 4. N=3 mice (out of N=9) for PHx representative mice (Fig. 1B) show no or little CD133 expression. Why are 33% of these mice do not show this expression since the other 6 mice do?

Answer 4: We think this is a natural result that reflects individual differences. As shown in Fig. 1A, the *Prom1* mRNA level at 48 hours after PHx increased significantly compared to sham, but the variability was very high. Such cases can easily be found in several previous studies¹⁶. To clarify upregulation of PROM1 protein at 48 hours after PHx, a plot of band intensity for Fig. 1B was added with individual values. Also in this plot, although the variability was too high, the PROM1 protein level after PHx was increased significantly compared to sham.

Minor Comments:

Comment 1. In Fig 2F, expression of DAPI and Ki67 should be enhanced as the frequency of Ki67 looks greater from sham model compared to PHx mice. IRS or some other form of quantification should be shown for at least n=3 mice for each IHC image.

Answer 1: As mentioned in the figure legend and in the "Materials and methods" section, the histogram counting Ki-67-expressing cells after PHx for Fig. 2F was obtained by statistical

analysis of 5 images per mouse (n=3) using ImageJ software (Fig. 2G). Fig 2F showed that there were no cells expressing Ki-67 in the *Prom1^{ff}* and *Prom1^{LKO}* sham liver. Since the background signal of the sham images seemed to cause confusion, the sham images were replaced in the revised version.

Comment 2. mRNA expression levels from Figure 1A shows expression after 24 and 120 hours after PHx. It would be good to see Western blot representative blots of at least n=3 for each of these groups as only 48 hours after PHx was shown (Fig. 1B).

Answer 2: At the reviewer's request, immunoblots for PROM1 in 24h and 120h PHx livers (n=4 for each group) were added in below Figure R4.

Figure R4. PROM1 protein level is increased in liver 24 hours and 120 hours after partial hepatectomy. Immunoblotting for PROM1 in wild type livers 24 hours and 120 hours after PHx (N=2 for sham, N=4 for 24 and 120 hours PHx).

Reviewer #3 (Remarks to the Author):

The authors of the manuscript entitled “Central role of Prominin-1 in lipid rafts during liver regeneration” describe the implication of the stem cell marker prominin-1 and lipid rafts during the process of liver regeneration, a phenomenon that relies on the interaction of the N-terminal domain of prominin-1 with the interleukin-6 signal transducer glycoprotein 130. Although this manuscript presents interesting data, many major concerns need to be clarified, including the role of lipid rafts in these processes.

Major Points

Comment 1. Please provide a high power view of the immunofluorescence images to appreciate the subcellular localization of prominin-1 in the cells of interest and describe in detail its

subcellular localization. For example, where is it located in the hepatocytes of the PHx liver? Immunogold electron microscopy could help define its localization. As mentioned in the introduction, prominin-1 is found in membrane protrusions. Please provide data on prominin-1 localization in primary hepatocytes under the different conditions used.

Answer 1. To clarify the subcellular localization of PROM1, PROM1 immunofluorescence signals were observed in *Prom1^{ff}* or *Prom1^{LKO}* livers after sham or PHx at higher magnification (100X). As shown in below Figure R5, PROM1 was expressed in the plasma membrane of ductal cells and hepatocytes in *Prom1^{ff}* livers, but not in *Prom1^{LKO}* livers. In our previous report¹⁰, we confirmed that PROM1 was expressed in microvilli of primary hepatocytes using an electron microscopy. Thus, we tried to confirm the localization of PROM1 in the microvilli of hepatocytes in PHx liver by correlative light and electron microscopy (CLEM) as previously described¹⁰. As shown in below Figure R6, immunogold-labeled PROM1 was localized in the microvilli of hepatocytes in *Prom1^{ff}* liver, but not in *Prom1^{LKO}* liver. These data showed that PROM1 was localized in the microvilli of hepatocytes from PHx liver.

Figure R5. PROM1 is expressed in the plasma membrane of ductal cells and hepatocytes in livers. PROM1 immunofluorescence in sham and PHx livers. Upper panels showed PROM1 expression in ductal cells (D). Lower panels showed PROM1 expression in hepatocytes (H). Scale bar = 10um.

Figure R6. PROM1 is localized in the microvilli of hepatocytes in PHx livers. Correlative light electron microscopy (CLEM) showed PROM1 expression in the microvilli of hepatocytes in PHx livers. PROM1 was labeled with a monoclonal rat antibody (13A4) and was visualized using an Alexa Fluor 488-Fluoro Nanogold (Nanoprobes). The left panels showed the

immunofluorescence signals of PROM1 in PHx liver sections using a confocal microscopy (Scale bar = 50um). The right panels showed the subcellular localization of immuno-gold labeled PROM1 in the sections (Scale bar = 0.5um). The yellow dotted boxes were enlarged.

Comment 2. Do prominin-1 knockout mice show a phenotype related to a liver defect? This information must be provided.

Answer 2. We added a paragraph for liver phenotypes of PROM1-deficient mice in the “Discussion” section as the followings (References were cited in the main text):

“Indeed, PROM1-deficiency leads to the prevention of glucagon-induced gluconeogenesis via inactivating the function of radixin as A kinase-anchoring protein (AKAP), and aggravation of bile duct ligation (BDL)-induced liver fibrosis via SMAD7 degradation, indicating that PROM1 has different functions in the liver.”

Comment 3. In the Results or Methods sections, the authors should explain the use of Brij-35 as a detergent employed for the isolation of lipid rafts. Classically, Triton X-100 is used. Indeed, GP130 is insoluble in Triton X-100 (Ref. 19), whereas prominin-1 is soluble in Triton X-100, suggesting that they do not localize in the same type of lipid rafts (Ref. 9). The authors should comment on this, and perhaps address these questions by comparing with data using Triton X-100 instead of Brij-35. Was co-localization of prominin-1 and GP130 observed under the different conditions used, and with the different prominin-1 deletion mutants?

Answer 3. As you mentioned, in Ref. 19 (in previous version of the manuscript)¹⁴, GP130 is insoluble in TX-100, but in another study, GP130 is mainly soluble in TX-100¹³. To clarify GP130 solubility in TX-100, we isolated lipid rafts using TX-100 (below Figure. R7) or Brij-35 (Fig. 5B in main figures) in PHx livers. GP130 was soluble in TX-100 but insoluble in Brij-35 from PHx *Prom1^{fl/fl}* livers. However, PROM1 deficiency leads to solubilize GP130 in Brij-35.

To further confirm the solubility of GP130 according to detergent, we investigated detergent insolubility using Brij-35 and TX-100 in HEK 293 cells after overexpressing

different PROM1 deletion mutants. Detergent soluble (S, supernatant) and insoluble (P, pellet) fractions were obtained from HEK 293 whole cell lysates after centrifugation at 17,000 g for 10 min. In the absence of PROM1, GP130 was mainly soluble in both TX-100 and Brij-35 (below Figure R8). The overexpression of PROM1 and other deletion mutants, along with Brij-35-insoluble PROM1 and other mutants, increased Brij-35-insoluble GP130 but not TX-100-insoluble GP130. Alternatively, PROM1^{GPI-EX1} overexpression increased TX-100 and Brij-35-insoluble GP130. These data indicate that the upregulated PROM1 recruits GP130 from non-rafts to Brij-35 resistant rafts.

Figure R7. GP130 is soluble in TX-100 in the liver. Detergent-resistant lipid rafts were isolated from *Prom1^{f/f}* and *Prom1^{LKO}* male mouse livers using a TX-100 48 hours after sham and PHx. Protein expression levels of PROM1, GP130, and Flotillin-1 were determined by immunoblotting in each fraction after sucrose gradient ultracentrifugation.

Figure R8. PROM1 and PROM1 deletion mutants increase Brij-35-insoluble GP130 but not TX-100-insoluble GP130. Detergent soluble (S, supernatant) and insoluble (P, pellet) fractions were obtained from HEK 293 cells overexpressing GP130-His and FLAG-tagged different PROM1 mutants after centrifugation at 17,000 g for 10 min. GP130 and PROM1 expression determined by immunoblotting for His and FLAG.

Comment 4. Is the interaction of prominin-1 and GP130 dependent on lipid rafts? According to the methods, the materials prepared for immunoisolation experiments are based on detergent lysates obtained after cell solubilization with the detergent NP-40, which will completely solubilize prominin-1, suggesting that lipid rafts are not involved in this cross-talk. This should be clarified. Does membrane cholesterol play a role in these interactions?

Answer 4. Agreeing with reviewer's comment, we investigated the interaction of PROM1 and GP130 after membrane cholesterol depletion. After methyl-beta-cyclodextrin (M β CD) treatment, whole cell lysates were obtained using NP-40 or Brij-35 from HEK 293 cells overexpressing PROM1-FLAG and GP130-His. Cholesterol depletion was confirmed by PROM1 and GP130 solubility in Brij-35 (below Figure R9). Coimmunoprecipitation experiments using NP-40 lysates showed that the interaction between PROM1 and GP130 was not changed by M β CD treatment (below Figure R9). These data suggested that the interaction between PROM1 and GP130 was not involved in lipid rafts.

Figure R9. The molecular interaction between PROM1 and GP130 is not limited in lipid rafts. The molecular interaction between PROM1 and GP130 was determined by co-immunoprecipitation from HEK 293 cells overexpressing PROM1-FLAG and GP130-His after 10mM MβCD treatment for 1 hour. The membrane cholesterol depletion was confirmed by immunoblotting for PROM1-FLAG and GP130-His from Brij-35 soluble (supernatant, S) and insoluble (pellet, P) fractions (Right panel). Co-immunoprecipitation was performed from NP-40 lysates (Left panel). WCL, whole cell lysates; IP, immunoprecipitation; IgG, normal IgG.

Comment 5. Do prominin-1 and GPI-anchored PROM1-EX1 co-localize in the same type of lipid rafts? According to Ref. 9, they will not. This question needs to be addressed biochemically and with respect to their co-localization in the cells of interest.

Answer 5. There are two types of lipid rafts according to Ref. 9 (in previous version of the manuscript)⁴. GPI-anchored proteins such as alkaline phosphatase localizes in TX-100-insoluble rafts. PROM1, which is co-localized with WGA, localizes in Brij-35-insoluble rafts. Thus, PROM1 is not co-localized with alkaline phosphatase. Thus, we determined whether PROM1 and PROM1^{GPI-EX1} are present in different lipid rafts. As shown in Figure R8, PROM1 was soluble in TX-100, but PROM1^{GPI-EX1} was insoluble in TX-100 in HEK 293 cells.

To further address this question, cell-surface immunostaining for PROM1 and PROM1^{GPI-EX1} was performed in HEK 293 cells overexpressing either or both of untagged PROM1 and FLAG-tagged PROM1^{GPI-EX1}. PROM1 was labeled with an anti-PROM1 antibody that binds to EX2 and EX3 whereas PROM1^{GPI-EX1} was labeled with an anti-FLAG antibody. As shown in below Figure R10, PROM1 and PROM1^{GPI-EX1} were not co-localized with each other (Fig. R10A). In addition, WGA was co-localized with PROM1 but not with PROM1^{GPI-EX1} (Fig. R10B). All these data suggested that PROM1 and PROM1^{GPI-EX1} were not co-localized in the same type of lipid rafts, consistent with reviewer's comment.

Figure R10. PROM1 and PROM1^{GPI-EX1} is not co-localized in the same type of lipid rafts. HEK 293 cells were transfected with untagged PROM1 and/or FLAG-tagged PROM1^{GPI-EX1} for 24 hours. (A) Double surface immunofluorescence for PROM1 and FLAG. (B) The surface immunofluorescence for PROM1 or FLAG after WGA staining. Scale bar=10um.

Other Points

Comment 1: In the Introduction section, the authors should describe the normal expression pattern of prominin-1 in liver (human and mouse) including canals of Hering and bile ducts (e.g., Immervoll et al. BMC Cancer 2008 PMID: 18261235; Karbanová et al. J Histochem Cytochem 2008 PMID: 18645205). These papers should be cited instead of the review of Glumac and Lebeau (Ref. 11) which does not provide such detailed information. It is essential

to give credit to genuine contributors. Citation of manuscripts that have not yet been peer-reviewed should not be encouraged, as is the case in bioRxiv (e.g., Ref. 16).

Answer 1: At the reviewer's request, we added the information on the PROM1 expression pattern in liver with Ref. 24, 25 (in revised version of the manuscript)^{17, 18} in Introduction section as the followings: "Specifically, PROM1 has been known to express in the liver including canals of Hering, bile ducts and hepatocytes."

To describe PROM1 as a cancer stem cell marker, Ref. 17-20 (in revised version of the manuscript)^{19, 20, 21, 22} were cited instead of Ref. 11 (in previous version of the manuscript)²³ in the revised manuscript. Because Ref. 16 (in previous version of the manuscript)²⁴ paper is being published in Exp. Mol. Med., we would like to maintain the reference.

Comment 2: In Supplementary Table 2, authors should provide the clone name and/or catalog number of the antibody. A reference for each must be provided as validation.

Answer 2: At the reviewer's request, we have provided the clone name and/or catalog number of the antibodies in the revised version.

Comment 3: In the Methods section (Immunofluorescence staining), is the rat anti-PROM1 (Thermo Fisher Scientific) the same as described in Supplementary Table 2 and acquired from eBioscience? Are we talking about the rat monoclonal antibody 13A4 used to identify prominin-1 (Weigmann et al. PNAS 1997 PMID: 9356465)? These details are important given that anti-prominin-1 antibodies have been the subject of much controversy.

Answer 3: The rat anti-PROM1 (Thermo Fisher scientific) in the "Materials and methods" section is same one presented in Supplementary Table 2 (ebioscience). We used the rat monoclonal antibody (13A4) to detect PROM1 expression in mice liver by immunoblotting and immunofluorescence. In normal liver, PROM1 expression was too low, so it was challenging to detect by immunoblotting, but in PHx liver, because PROM1 was upregulated, it was easily detected. Because the immunoreactive bands were not detected in the *Prom1*^{LKO} liver lysates, the observed bands were specific for PROM1 (Fig. 2C, 3D, 4A, C)

Determining the tissue distribution and subcellular localization of PROM1 is difficult because of the limited immunoreactivity of glycosylated PROM1. To overcome this limitation, we modified Shmelkov's method²⁵ to visualize PROM1 by immunofluorescence analysis as

described in "Materials and methods" section. This method enables us to demonstrate that PROM1 was localized in the microvilli and the plasma membrane of hepatocytes (Fig. 1C, D and Fig. R5, R6). Because the signal was not detected in the *Prom1*^{LKO} liver, the observed immunofluorescence signal was specific for PROM1 (Fig. R5, R6).

Comment 4: In the histograms, please provide individual values and not just the mean and S.E.M.

Answer 4: At the reviewer's request, we added individual values for all plots in the revised figure.

Comment 5: The uncropped blots should be provided in the Supplementary Materials. Does any other prominin-1-related immunoreactive bands appear on the whole blot (e.g., Figure 2C)? Molecular mass marker should be indicated.

Answer 5: The uncropped blots for all western blot data were provided in "Source Data". Molecular mass marker was also indicated on the uncropped blots. In the case of blots for PROM1 in Fig.2C mentioned by reviewer, since the quality of previous data was not great, it was replaced with re-blotted bands of the same samples. No other immunoreactive bands appeared in replaced data.

Comment 6: What are the different lanes of the blot (PHx) shown in Figure 1B? Different independent samples? The variability seems to be very high. How do the authors explain this?

Answer 6: Fig. 1B shows immunoblots for PROM1 in wild type sham liver (N=2) and 48h PHx liver (N=9). Left two lanes represent sham liver samples and the next nine lanes represent 48h PHx liver specimen of 8-week-old wild-type male mice. Each different lane corresponds to a different mouse liver specimen. To avoid confusion, we have marked the number of mice above the lanes in Fig.1B.

We think this is a natural result that reflects individual differences. As shown in Fig. 1A, the *Prom1* mRNA level at 48 hours after PHx increased significantly compared to sham, but the variability was very high. Such cases can easily be found in several previous studies¹⁶.

To clarify upregulation of PROM1 protein at 48 hours after PHx, a plot of band intensity for Fig. 1B was added with individual values. Also in this plot, although the variability was too high, the PROM1 protein level after PHx was increased significantly compared to sham.

Comment 7: Does apoptosis occur under the different conditions?

Answer 7: To investigate whether apoptosis occur under the different conditions, we performed TUNEL assay in PHx and CCl₄ model. As shown in Fig. S1 and below Figure R11, apoptosis occurred after CCl₄ injection but not after PHx. Fibrotic liver specimen from BDL-operated mouse was used as a positive control. The difference in CCl₄-induced apoptosis between *Prom1^{fl/fl}* and *Prom1^{LKO}* livers was not significant (Fig. S1). Our PHx data is consistent with reports that apoptosis is prevented during normal liver regeneration after PHx²⁶.

Figure R11. Apoptosis does not occur after partial hepatectomy. Apoptosis was analyzed in 8-week-old male *Prom1^{fl/fl}* and *Prom1^{LKO}* mice livers by TUNEL assay 48 hours after PHx. BDL liver specimen was used as a positive control. Scale bar = 50 μ m. TUNEL, terminal deoxynucleotidyl transferase dUTP nick end labeling; BDL, bile duct ligation.

Comment 8: Figure 4A, C, as an internal control, prominin-1 blots should be shown.

Answer 8: PROM1 immunoblots were added to each panel (Fig. 4A, C).

Comment 9: It is indicated in the Methods that variant 2 of the human PROM1 transcript is used. Please provide the source of this nomenclature. According to Fargeas et al. (Tissue Antigens 2007, PMID:17498271), the human s2 variant will be 865 amino acids in length, not 856 residues as shown in Figure 6A. The length of 856 amino acids will correspond to the s1 variant.

Answer 9: We used human PROM1 transcript variant 2 (GenBank: NM_001145847.1), also known as “s1” splice variant. The variant 2 (s1) encodes “isoform 2” and isoform 2 is known by the synonyms “AC133-2 and S1” (Uniprot: O43490-2) and corresponds to 856 amino acids. In a previous report, because AC133-2 is highly expressed in the human liver²⁷, we used this variant.

Reference

1. Levental KR, Levental I. Isolation of giant plasma membrane vesicles for evaluation of plasma membrane structure and protein partitioning. *Methods Mol Biol* **1232**, 65-77 (2015).
2. Sezgin E, Kaiser HJ, Baumgart T, Schwille P, Simons K, Levental I. Elucidating membrane structure and protein behavior using giant plasma membrane vesicles. *Nat Protoc* **7**, 1042-1051 (2012).
3. Monsigny M, Roche AC, Sene C, Maget-Dana R, Delmotte F. Sugar-lectin interactions: how does wheat-germ agglutinin bind sialoglycoconjugates? *Eur J Biochem* **104**, 147-153 (1980).
4. Roper K, Corbeil D, Huttner WB. Retention of prominin in microvilli reveals distinct cholesterol-based lipid micro-domains in the apical plasma membrane. *Nat Cell Biol* **2**, 582-592 (2000).
5. Thomas S, Preda-Pais A, Casares S, Brumeanu TD. Analysis of lipid rafts in T cells. *Mol Immunol* **41**, 399-409 (2004).
6. Thomas S, Kumar RS, Brumeanu TD. Role of lipid rafts in T cells. *Arch Immunol Ther Exp (Warsz)* **52**, 215-224 (2004).
7. Korade Z, Kenworthy AK. Lipid rafts, cholesterol, and the brain. *Neuropharmacology* **55**, 1265-1273 (2008).

8. Levental I, Byfield FJ, Chowdhury P, Gai F, Baumgart T, Janmey PA. Cholesterol-dependent phase separation in cell-derived giant plasma-membrane vesicles. *Biochem J* **424**, 163-167 (2009).
9. Castello-Serrano I, Lorent JH, Ippolito R, Levental KR, Levental I. Myelin-Associated MAL and PLP Are Unusual among Multipass Transmembrane Proteins in Preferring Ordered Membrane Domains. *J Phys Chem B* **124**, 5930-5939 (2020).
10. Lee H, *et al.* Prominin-1-Radixin axis controls hepatic gluconeogenesis by regulating PKA activity. *EMBO Rep* **21**, e49416 (2020).
11. Port MD, Gibson RM, Nathanson NM. Differential stimulation-induced receptor localization in lipid rafts for interleukin-6 family cytokines signaling through the gp130/leukemia inhibitory factor receptor complex. *J Neurochem* **101**, 782-793 (2007).
12. Yanagisawa M, Nakamura K, Taga T. Roles of lipid rafts in integrin-dependent adhesion and gp130 signalling pathway in mouse embryonic neural precursor cells. *Genes Cells* **9**, 801-809 (2004).
13. Buk DM, Waibel M, Braig C, Martens AS, Heinrich PC, Graeve L. Polarity and lipid raft association of the components of the ciliary neurotrophic factor receptor complex in Madin-Darby canine kidney cells. *J Cell Sci* **117**, 2063-2075 (2004).
14. Sehgal PB, Guo GG, Shah M, Kumar V, Patel K. Cytokine signaling: STATS in plasma membrane rafts. *J Biol Chem* **277**, 12067-12074 (2002).
15. Kim KB, Lee JS, Ko YG. The isolation of detergent-resistant lipid rafts for two-dimensional electrophoresis. *Methods Mol Biol* **424**, 413-422 (2008).
16. Pritchard CC, Hsu L, Delrow J, Nelson PS. Project normal: defining normal variance in mouse gene expression. *Proc Natl Acad Sci U S A* **98**, 13266-13271 (2001).
17. Immervoll H, Hoem D, Sakariassen PO, Steffensen OJ, Molven A. Expression of the "stem cell

- marker" CD133 in pancreas and pancreatic ductal adenocarcinomas. *BMC Cancer* **8**, 48 (2008).
18. Karbanova J, *et al.* The stem cell marker CD133 (Prominin-1) is expressed in various human glandular epithelia. *J Histochem Cytochem* **56**, 977-993 (2008).
 19. Liu G, *et al.* Analysis of gene expression and chemoresistance of CD133+ cancer stem cells in glioblastoma. *Mol Cancer* **5**, 67 (2006).
 20. Bao S, *et al.* Glioma stem cells promote radioresistance by preferential activation of the DNA damage response. *Nature* **444**, 756-760 (2006).
 21. Florek M, *et al.* Prominin-1/CD133, a neural and hematopoietic stem cell marker, is expressed in adult human differentiated cells and certain types of kidney cancer. *Cell Tissue Res* **319**, 15-26 (2005).
 22. Yin S, *et al.* CD133 positive hepatocellular carcinoma cells possess high capacity for tumorigenicity. *Int J Cancer* **120**, 1444-1450 (2007).
 23. Glumac PM, LeBeau AM. The role of CD133 in cancer: a concise review. *Clin Transl Med* **7**, 18 (2018).
 24. Lee H, *et al.* Hepatocytic Prominin-1 protects against liver fibrosis by stabilizing the SMAD7 protein. *bioRxiv*, 846493 (2019).
 25. Shmelkov SV, *et al.* CD133 expression is not restricted to stem cells, and both CD133+ and CD133- metastatic colon cancer cells initiate tumors. *J Clin Invest* **118**, 2111-2120 (2008).
 26. Iimuro Y, *et al.* NFkappaB prevents apoptosis and liver dysfunction during liver regeneration. *J Clin Invest* **101**, 802-811 (1998).
 27. Yu Y, Flint A, Dvorin EL, Bischoff J. AC133-2, a novel isoform of human AC133 stem cell antigen. *J Biol Chem* **277**, 20711-20716 (2002).

REVIEWERS' COMMENTS

Reviewer #1 (Remarks to the Author):

The authors have made a considerable effort to address my concerns and I am satisfied with the revisions. I do however have one final comment. I personally found the reviewer figures to be very helpful. (This applies to all of them, not just the ones in response to my comments.) I suspect the questions they address might be shared by other readers of the manuscript. The authors may thus want to consider adding these data to the manuscript as supplementary figures. However, this is ultimately up to the authors and editors to decide.

Reviewer #2 (Remarks to the Author):

Authors faithfully performed many additional experiments to rectify almost all concerns that reviewers raised.

In Figure 6B and 6C, IP-Western blots were performed. However, isotype matched antibody IP experiments are missing. This negative control would support to exclude whether the detected bands are not heavy and/or light chains of immunoprecipitation antibody.

Figure 5G immunofluorescence data to show the colocalization of PROM21 and GRP130 are convincing.

Overall, authors faithfully rectified reviewers' concerns to support the author's conclusions.

Reviewer #3 (Remarks to the Author):

The authors of the manuscript entitled "Central role of Prominin-1 in lipid rafts during liver regeneration" have addressed my main concerns, but to be fully satisfactory the new data provided in the rebuttal letter, including Fig. R6: Correlative light electron microscopy; Fig. R9: Molecular interaction between PROM1 and GP130 is not restricted to lipid rafts; and Fig. R10: PROM1 and PROM1GPI-EX1 are not co-localized in the same type of lipid rafts, should be included in the manuscript, at least as Supplementary information. Of course, the text must be adapted accordingly. This will strengthen the manuscript, especially for Nature Communication.

Title: Central role of Prominin-1 in lipid rafts during liver regeneration

Manuscript Number: #NCOMMS-21-51136A

Authors: Myeong-Suk Bahn, Dong-Min Yu, Myoungwoo Lee, Sung-Je Jo, Ji-Won Lee, Ho-Chul Kim, Hyun Lee, Hong Lim Kim, Arum Kim, Jeong-Ho Hong, Jun Seok Kim, Seung-Hoi Koo, Jae-Seon Lee, and Young-Gyu Ko

Reviewer #1

The authors have made a considerable effort to address my concerns and I am satisfied with the revisions. I do however have one final comment. I personally found the reviewer figures to be very helpful. (This applies to all of them, not just the ones in response to my comments.) I suspect the questions they address might be shared by other readers of the manuscript. The authors may thus want to consider adding these data to the manuscript as supplementary figures. However, this is ultimately up to the authors and editors to decide.

Answer: We added reviewer-requested data in Supplementary figures.

Reviewer #2

Authors faithfully performed many additional experiments to rectify almost all concerns that reviewers raised.

In Figure 6B and 6C, IP-Western blots were performed. However, isotype matched antibody IP experiments are missing. This negative control would support to exclude whether the detected bands are not heavy and/or light chains of immunoprecipitation antibody.

Figure 5G immunofluorescence data to show the colocalization of PROM1 and GP130 are convincing.

Overall, authors faithfully rectified reviewers' concerns to support the author's conclusions.

Answer: We repeated IP experiments with negative controls. The results are shown in Fig. 6B and C of revised manuscript.

Reviewer #3 (Remarks to the Author):

The authors of the manuscript entitled "Central role of Prominin-1 in lipid rafts during liver regeneration" have addressed my main concerns, but to be fully satisfactory the new data provided in the rebuttal letter, including Fig. R6: Correlative light electron microscopy; Fig.

R9: Molecular interaction between PROM1 and GP130 is not restricted to lipid rafts; and Fig. R10: PROM1 and PROM1GPI-EX1 are not co-localized in the same type of lipid rafts, should be included in the manuscript, at least as Supplementary information. Of course, the text must be adapted accordingly. This will strengthen the manuscript, especially for Nature Communication.

Answer: We added reviewer-requested data in Supplementary Figures.